# PREFERENTIAL DYNAMIC MODELING WITH FORWARD-BACKWARD SMOOTHING

## ABSTRACT

Estimating a secondary signal (e.g., behavior) from neural activity over time is central to both causal online decoding and non-causal offline inference in neuroscience. Existing two-signal latent state-space modeling methods typically either support causal prediction of the secondary signal from the primary signal or non-causal inference (smoothing), but rarely both; here we extend one analytical linear method (PSID) and one nonlinear deep learning method (DPAD) beyond causal prediction to also support non-causal inference, yielding a more universally applicable family of methods. We provide theoretical derivations extending PSID to enable optimal filtering and optimal smoothing of the secondary signal. We show that, in the PSID setting, the presence of a secondary signal increases identifiability. This allows us to uniquely learn the quantities needed for the optimal Kalman update via a reduced-rank regression step that augments the standard SVD-based PSID algorithm, yielding our first contribution, *PSID with filtering*. We next design a forward-backward construction for smoothing, yielding our second contribution, *PSID with smoothing*. For nonlinear prioritized modeling, we extend DPAD to a bidirectional variant that combines forward and backward hidden states at readout to perform smoothing, yielding our third contribution, *DPAD with smoothing*. In simulations, we validate that PSID with filtering and smoothing reach ideal performance. In non-human primate motor cortex data, PSID with smoothing consistently improves over PSID with filtering, which improves over one-step-ahead prediction with standard PSID. Finally, we test DPAD with smoothing on three Neural Latents Benchmark (NLB) datasets, where it achieves the top behavior-decoding result on at least one dataset and near-top performance in behavior decoding and held-out neural prediction on all three. Together, these methods form a family with wide-ranging applications, from causal online decoding to offline inference, in both linear and nonlinear settings.

## 1 INTRODUCTION

Understanding the relationship between neural activity and behavior is a critical goal in neuroscience and neurotechnology. A key challenge is accurate estimation of a secondary signal, such as behavior, from neural activity over time. While system identification methods provide principled approaches to model the temporal dynamics of neural signals, estimating a secondary signal from neural observations presents unique challenges that depend on the temporal information available. Specifically, one-step-ahead prediction estimates the secondary signal at time $k$ given neural observations up to time $k-1$, filtering estimates it given observations up to the current time $k$, and smoothing estimates it given observations over an entire time window up to some future time $N$ (Appendix A.1). While these three estimation problems have well-established solutions for directly observable states, extending them to latent variable models with a secondary signal is nontrivial and has received limited attention.

There are fundamental differences between filtering and smoothing in two-signal system identification compared to one-signal scenarios. In the one-signal scenario, which is tackled by methods such as subspace identification (SID) (Van Overschee & De Moor, 1996), filtered or smoothed predictions of the observed signal itself are not of interest, because the sample to be predicted is already observed, making the observation its own optimal estimate (under standard Gaussian state space model assumptions). When such smoothed predictions are of interest in the one-signal setting, e.g., in LFADS (Pandarinath et al., 2018), they often rely on inductive biases that aim to estimate an

underlying latent process rather than the specific observed signal. In contrast, in two-signal settings where a secondary signal such as behavior or held out neural activity must be estimated from primary neural observations (tackled by methods such as PSID (Sani et al., 2021), DPAD (Sani et al., 2024), TNDM (?), or Ctrl-TNDM (Kudryashova et al., 2025)), filtering and smoothing become interesting even at the level of predicting the exact realized secondary signal. However, no existing method aims to address all three types of estimation problems in the two-signal setting.

While PSID (Sani et al., 2021) and subsequently DPAD (Sani et al., 2024) have aimed to achieve optimal causal prediction of the secondary signal (i.e., one-step-ahead prediction), neither aims to support optimal filtering (same-step prediction) or smoothing (all-time prediction). On the other hand, while TNDM (?) and Ctrl-TNDM (or BAND) (Kudryashova et al., 2025) address the smoothing problem, they are based on sequential autoencoders, which rely on observing all trial data to estimate the initial latent state and thus fundamentally do not support prediction or filtering. Our work bridges this gap by: (i) analytically extending PSID to derive optimal filtering and smoothing estimators for the secondary signal, and (ii) extending DPAD to support smoothing.

To enable filtering, we derive a Kalman-style update step yielding the optimal filtered estimate through a new gain parameter learned via reduced-rank regression. Building upon this filtering framework, we extend PSID to support smoothing via a forward-backward design in which the backward model captures the residual of the secondary signal not predicted by the forward model. To extend DPAD to support smoothing, we implement a similar forward-backward design using a bidirectional RNN architecture that combines forward and backward hidden states at the readout layer, enabling non-causal inference (smoothing). We discuss how these extensions relate to existing methods in Section 2.

We provide derivations for the extensions of PSID and empirically validate them through extensive simulations and in real data. In simulations, our extended PSID method reaches ideal filtering and smoothing performance, matching that of the true model parameters. We further evaluate extended PSID on real non-human primate motor cortical data (O'Doherty et al., 2020), where again filtering and smoothing yield substantial improvements in behavior decoding accuracy. Finally, we validate our extended DPAD with smoothing on the Neural Latents Benchmark (NLB), where it achieves top or near-top performance in both behavior decoding and held-out neural prediction across multiple datasets, establishing competitive results with leading methods on the leaderboard, including Ctrl-TNDM, AutoLFADS, and LangevinFlow (Section 4.5).

Collectively, our contributions provide a comprehensive framework addressing all three estimation problems (prediction, filtering, and smoothing) through both linear analytical modeling (PSID) and nonlinear numerical optimization (DPAD). This coverage enables applications ranging from real-time analyses, where causality is critical (prediction and filtering), to offline analyses, where non-causal smoothing can improve accuracy. It also spans settings from embedded-system or controller design, where linear dynamical modeling may be preferred, to brain-computer interface applications, where flexible nonlinear modeling may be preferred as long as it yields more accurate decoding.

## 2 RELATED WORK

Our work addresses the intersection of several methodological challenges, making it partially related to multiple prior methods: *1)* PSID (Sani et al., 2021): addresses prioritized learning for two-signal system identification but only targets one-step-ahead prediction. By extending PSID to include filtering and smoothing, we address this limitation. *2)* DPAD (Sani et al., 2024): extends PSID's dissociation and prioritization of shared dynamics to nonlinear modeling, but only targets prediction. Our extension of DPAD to support smoothing builds on that method. *3)* Linear NDM: traditional one-signal neural dynamic modeling (NDM) methods such as subspace identification (Van Overschee & De Moor, 1996), which are a special case of PSID and a baseline for it in Sani et al. (2021). *4)* LFADS (Pandarinath et al., 2018): a one-signal non-causal method based on sequential autoencoder that aims to infer single-trial latent neural state evolutions. *5)* AutoLFADS (?): An extension of LFADS that uses population-based training (Jaderberg et al., 2017) for hyperparameter optimization, leading to better results. *6)* TNDM (?): A two-signal extension of LFADS that dissociates behaviorally relevant dynamics into separate latent states. *7)* Ctrl-TNDM (or BAND) (Kudryashova et al., 2025): A two-signal extension of TNDM that aligns behavior and neural dynamics via a prior over the behavior latent state.

## 3 METHODS

### 3.1 MODEL FORMULATION FOR LINEAR METHODS

We model the temporal dynamics of two time series $\boldsymbol{y}_k \in \mathbb{R}^{n_y}$ and $\boldsymbol{z}_k \in \mathbb{R}^{n_z}$ in terms of the latent state $\boldsymbol{x}_k^s \in \mathbb{R}^{n_x}$ as:

$$\begin{cases} \boldsymbol{x}_{k+1}^s &= A \quad \boldsymbol{x}_k^s \quad + \quad \boldsymbol{w}_k \\ \boldsymbol{y}_k &= C_y \quad \boldsymbol{x}_k^s \quad + \quad \boldsymbol{v}_k \\ \boldsymbol{z}_k &= C_z \quad \boldsymbol{x}_k^s \quad + \quad \boldsymbol{\epsilon}_k \end{cases} \tag{1}$$

where $\boldsymbol{w}_k \in \mathbb{R}^{n_x}$ and $\boldsymbol{v}_k \in \mathbb{R}^{n_y}$ are white Gaussian noises with the following cross-correlation:

$$\mathbb{E}\left\{ \begin{bmatrix} \boldsymbol{w}_k \\ \boldsymbol{v}_k \end{bmatrix} \begin{bmatrix} \boldsymbol{w}_k \\ \boldsymbol{v}_k \end{bmatrix}^T \right\} = \begin{bmatrix} Q & S \\ S^T & R \end{bmatrix}. \tag{2}$$

$\boldsymbol{\epsilon}_k \in \mathbb{R}^{n_z}$ is a general random process representing the dynamics of $\boldsymbol{z}_k$ that are not encoded in $\boldsymbol{y}_k$. As such, we only assume that $\boldsymbol{\epsilon}_k$ is zero-mean and independent of $\boldsymbol{x}_k^s$ and the other noises ($\boldsymbol{w}_k$ and $\boldsymbol{v}_k$). This is known as the stochastic form formulation for a latent state space model (Appendix A.3).

### 3.2 KALMAN FILTERING AND THE PREDICTOR FORM

Given $(A, C_y, Q, R, S)$, the Kalman filter computes the minimum-variance state estimates using a causal prediction-update recursion. For a thorough formulation of the Kalman filter, please see Appendix A.2 (equations 12-13). A compact recursive formulation that combines the prediction and update steps can be written as:

$$\hat{\boldsymbol{x}}_{k+1|k} = (A - KC_y)\hat{\boldsymbol{x}}_{k|k-1} + K\boldsymbol{y}_k \tag{3}$$

where $K$ is the Kalman gain, and $\hat{\boldsymbol{x}}_{k|k-1}$ is the Kalman predicted state at time $k$, given all prior observations $\boldsymbol{y}_{1:k-1}$. It is important to note that the total Kalman gain $K$ is a combination of two different gains for the prediction and update steps, i.e., $K \triangleq AK_f + K_v$. As long as only the one-step ahead prediction $\hat{\boldsymbol{x}}_{k|k-1}$ is of interest, we do not need to know the individual gains $K_f$ and $K_v$, as can be seen from equation 3. However, if filtering ($\hat{\boldsymbol{x}}_{k|k}$) is of interest, we need to know the individual gains $K_f$ and $K_v$, which as we will explain has ramifications in terms of system identification.

This formulation is also the basis for an alternative but equivalent formulation for the state-space model from equation 1, known as the predictor form, which is given by:

$$\begin{cases} \boldsymbol{x}_{k+1} &= A \boldsymbol{x}_k \quad + \quad K(\boldsymbol{y}_k - C_y \boldsymbol{x}_k) \\ \boldsymbol{y}_k &= C_y \boldsymbol{x}_k \quad + \quad \boldsymbol{e}_k \end{cases} \tag{4}$$

where $\boldsymbol{x}_k \triangleq \hat{\boldsymbol{x}}_{k|k-1}$. Next, we will briefly review the key concepts related to our derivation; for details see Appendix A.3, Van Overschee & De Moor (1996), and Katayama (2006).

The mapping from stochastic to predictor form is unique (up to a similarity transform) and can be obtained via the Kalman filter; the reverse mapping is non-unique due to redundancy of $(Q, R, S)$ (see Faurre's theorem in Appendix A.3). Practically, this implies $(A, C_y, K, \Sigma_e)$ are uniquely identifiable from $\boldsymbol{y}$, while $(Q, R, S)$ are not. Critically, this is not a limitation of system identification, but rather a fundamental characteristic of the latent state-space model. That is, $(A, C_y, K, \Sigma_e)$ are all external characteristics of the system in the sense that they affect its measurable properties (e.g., 2nd order statistics of $\boldsymbol{y}$), but $(Q, R, S)$ are internal characteristics meaning that they do not uniquely affect the behavior of the system (see Appendix A.7). Of note, the total Kalman gain $K$ is uniquely identifiable, while the individual gains $K_f$ and $K_v$ are not. This has major implications for model identifiability and the distinction between optimal prediction and optimal filtering/smoothing.

### 3.3 KALMAN SMOOTHING

Kalman smoothing is the optimal estimation of the latent state $\boldsymbol{x}_k$ given *all* observations up to $N$ (Appendix A.5). Two equivalent formulations are relevant: (i) the RTS smoother (equations 31-32), which updates $\hat{\boldsymbol{x}}_{k|k}$ directly using a backward recursion (equation 31); and (ii) the forward-backward two-filter formulation (equations 33-36), which runs a filter on the observations $\boldsymbol{y}$ in the reverse direction (from $N$ to 1) and fuses its states with those of the standard forward Kalman filter via a

weighted average. Our formulation of PSID with smoothing is motivated by the forward-backward two-filter formulation. Critically, in both Kalman smoothing formulations, the smoother relies on the $K_f$ parameter, which as discussed in the previous section, is not uniquely identifiable in the one-signal setting. As we will discuss below, the identifiability changes when two signals are present, which forms the basis of our extensions of PSID for filtering and smoothing.

### 3.4 EXTENDING PSID

#### 3.4.1 A REVIEW OF PSID

Preferential Subspace Identification (PSID) is a system identification method designed to model the dynamics of two time series, $\boldsymbol{y}_k \in \mathbb{R}^{n_y}$ and $\boldsymbol{z}_k \in \mathbb{R}^{n_z}$, where $\boldsymbol{z}_k$ is not expected to be measured during inference. PSID models the dynamics of the primary signal $\boldsymbol{y}_k$ while prioritizing and dissociating dynamics that are relevant to the secondary signal $\boldsymbol{z}_k$. The formulation of PSID is similar to equation 1 (or its predictor form equivalent 4), except that the latent state is split into two parts: $\boldsymbol{x}_k = [\hat{\boldsymbol{x}}_k^{(1)}; \hat{\boldsymbol{x}}_k^{(2)}]$. PSID operates in two stages. In the first stage, dynamics that are shared between the two time series (i.e., captured by $\hat{\boldsymbol{x}}_k^{(1)}$) are extracted via a projection of future behavior $\boldsymbol{z}_k$ onto corresponding past neural activity $\boldsymbol{y}_k$. Any residual neural dynamics are explained using additional latent states $\hat{\boldsymbol{x}}_k^{(2)}$ in an optional second stage (details in Appendix A.8).

In a two-signal setting like in PSID, the presence of $\boldsymbol{z}$ breaks the equivalence among stochastic form models that describe $\boldsymbol{y}$. Specifically, unidentifiable parameters such as $K_f$ become partially identifiable to the extent that they affect the optimal estimation of the secondary signal $\boldsymbol{z}$ (Appendix A.7). This allows us to develop filtering and smoothing capabilities for PSID, which we discuss next.

#### 3.4.2 PSID WITH FILTERING

To derive PSID with optimal filtering, our key idea is to select the parameter $K_f$ among all possible solutions of system identification that yields the best filtered estimate of the secondary time series $\boldsymbol{z}_k$. In other words, we want to solve the following error minimization over all training samples

$$\arg \min_{K_f} \sum_k \|\boldsymbol{z}_k - \hat{\boldsymbol{z}}_k\|_2^2, \tag{5}$$

where $\hat{\boldsymbol{z}}_k = C_z \hat{\boldsymbol{x}}_{k|k}$, and $\hat{\boldsymbol{x}}_{k|k}$ is computed using $K_f$. Critically, obtaining $C_z K_f$ is sufficient for implementing the optimal filter for predicting $\boldsymbol{z}_k$ from $\boldsymbol{y}_k$, without the need to learn $K_f$ separately:

$$\hat{\boldsymbol{z}}_{k|k} = C_z \hat{\boldsymbol{x}}_{k|k} = C_z(\hat{\boldsymbol{x}}_{k|k-1} + K_f(\boldsymbol{y}_k - C_y \hat{\boldsymbol{x}}_{k|k-1})) = \hat{\boldsymbol{z}}_{k|k-1} + C_z K_f(\boldsymbol{y}_k - \hat{\boldsymbol{y}}_{k|k-1}). \tag{6}$$

Thus, we can instead solve for (see Theorem 4, in Appendix A.8.3)

$$\arg \min_{C_z K_f} \sum_k \left\| \tilde{\boldsymbol{z}}_{k|k-1} - C_z K_f \tilde{\boldsymbol{y}}_{k|k-1} \right\|_2^2, \quad \text{s.t.} \quad \text{rank}(C_z K_f) = \min(n_x, n_y, n_z) \tag{7}$$

where $\tilde{\boldsymbol{z}}_{k|k-1} \triangleq \boldsymbol{z}_k - \hat{\boldsymbol{z}}_{k|k-1}$ and $\tilde{\boldsymbol{y}}_{k|k-1} \triangleq \boldsymbol{y}_k - \hat{\boldsymbol{y}}_{k|k-1}$ are the one-step-ahead prediction errors. Of note, we use reduced rank regression or RRR (?) to solve the above optimization, subject to the provided rank condition, because it can be shown that the rank of $C_z K_f$ can at most be $min(n_x, n_y, n_z)$ (Figure 1a). We discuss a more general version of this optimization in Appendix A.8.2 and the exact learning of $K_f$ itself in Appendix A.8.4. As a cautionary baseline, simply shifting $\boldsymbol{z}$ in time during training is not equivalent to optimal filtering when $S \neq 0$ (Appendix A.15).

#### 3.4.3 PSID WITH SMOOTHING

Inspired by the two-filter formulation for Kalman smoothers, we recognize that PSID learns the optimal filter in one direction. To apply a smoother, we further need to learn the optimal filter to predict residual data in the opposite direction. In effect, we are learning the forward and backward filters of the forward-backward smoother separately and directly from data. It should be noted that the backward model learned in PSID smoothing is *different* from the backward representation of the stochastic model (equation 18), which is explained in Appendix A.4.

More concretely, PSID with smoothing proceeds as follows (Figure 1b):

1. Apply PSID with filtering, which, as explained in the previous section, consists of PSID for prediction plus reduced rank regression to learn $C_z K_f$.

2. Compute the filtered estimate of the secondary time series $\hat{z}_{k|k}$ using this learned model.

3. Subtract $\hat{z}_{k|k}$ from the secondary time series to find the residual secondary time series:

$$\tilde{z}_{k|k} = z_k - \hat{z}_{k|k}. \tag{8}$$

4. Apply PSID with filtering in the opposite time direction and on the residual secondary signal $\tilde{z}_{k|k}$ as our new secondary signal. This gives us a new model in the reverse time direction.

The final prediction from this smoothing PSID, i.e., $\hat{z}_{k|N}$, is the sum of the predictions from the forward and backward filters

$$\hat{z}_{k|N} = \hat{z}_{k|k} + \hat{\tilde{z}}_{k|k}, \tag{9}$$

where $\hat{z}_{k|k}$ and $\hat{\tilde{z}}_{k|k}$ are the forward and backward PSID filtered estimates for sample $k$, respectively (Figure 1b). This formulation resembles the forward-backward Kalman smoothing formulation (equation 36), in that the final prediction is a weighted sum of the forward and backward predictions. See Appendix A.8.5 for a sketch of the proof for why this yields optimal smoothed estimates.

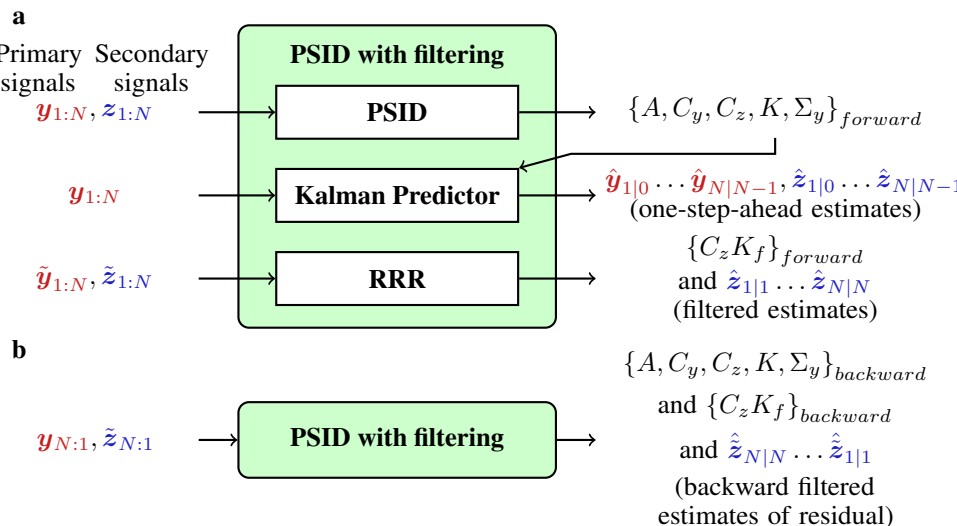

Figure 1: (**a**) Diagram of PSID with filtering. The method consists of three main steps: (1) Regular PSID learns the forward model parameters from input signals, (2) a Kalman predictor uses the learned model to make one-step-ahead predictions, and (3) Reduced Rank Regression (RRR) learns updated $C_z K_f$ parameters to produce optimal filtered estimates of the behavior signals. (**b**) Diagram of PSID with smoothing. The method first applies PSID with filtering as in (**a**), and obtains the error of the filtered estimate of the secondary signal, i.e., $\tilde{z}_{1:N}$. Next, $y$ and $\tilde{z}$ are reversed in time, i.e., $y_{N:1}$ and $\tilde{z}_{N:1}$, and passed to PSID with filtering to learn the parameter of the backwards model.

### 3.5 EXTENSION TO NONLINEAR MODELS: DPAD WITH SMOOTHING

We next apply our forward-backward smoothing architecture to an RNN-based nonlinear dynamical modeling method called DPAD (Dissociative Prioritized Analysis of Dynamics), by Sani et al. (2024). Like PSID, DPAD is a state-space model that prioritizes learning behaviorally relevant dynamics, but it does so via numerical optimization, which allows it to learn nonlinear mappings and latent recursions. The forward-pass inference model is given by

$$x_{k+1} = A'(x_k) + K(y_k) \quad ; \quad \hat{y}_k = C_y(x_k) \quad ; \quad \hat{z}_k = C_z(x_k), \tag{10}$$

where $A'$, $K$, $C_y$, and $C_z$ are nonlinear mappings implemented as multi-layer perceptrons (MLPs), in contrast to the analogous linear mappings in PSID. Notably, this formulation is a generalized form of the predictor form representation from equation 4 (where we define $A' \triangleq A - K C_y$), which means

that the model can readily be used for inference/prediction (Sani et al., 2024). As a special case, for example, if all parameters are linear, DPAD can numerically learn and implement an optimal Kalman predictor (Appendix A.9).

To incorporate smoothing, we implement a bidirectional RNN architecture. The model consists of a forward and a backward RNN that are trained simultaneously. The forward RNN processes the input time-series $\boldsymbol{y}_{1:N}$. The backward RNN is initialized with the final latent state of the forward RNN and processes the time-reversed input sequence $\boldsymbol{y}_{N:1}$. At each time step $k$, the latent states from the forward pass ($\boldsymbol{x}_k^{fwd}$) and the backward pass ($\boldsymbol{x}_k^{bwd}$) are concatenated to form a smoothed state, $\boldsymbol{x}_k^{smooth} = [\boldsymbol{x}_k^{fwd}; \boldsymbol{x}_k^{bwd}]$. This concatenated state is then used by readout networks $C_z$ and $C_y$ to predict the secondary or primary time series, respectively:

$$\hat{\boldsymbol{z}}_{k|N}^{\text{DPAD}} = C_z\Big([\,\boldsymbol{x}_k^{fwd},\,\boldsymbol{x}_k^{bwd}\,]\Big), \quad \hat{\boldsymbol{y}}_{k|N}^{\text{DPAD}} = C_y\Big([\,\boldsymbol{x}_k^{fwd},\,\boldsymbol{x}_k^{bwd}\,]\Big), \tag{11}$$

Overall, the training and inference procedure of DPAD with smoothing is as in the base DPAD, except given that the encoder RNNs are bidirectional, the final estimation is non-causal and aware of the complete sequence of primary time series and thus performs smoothing.

Architectural and optimization hyperparameters largely follow Sani et al. (2024), with the following changes: 1) We use bidirectional RNNs. 2) For held-out neural prediction, which has a different target distribution than behavior decoding, we treated held-out neural activity as the secondary signal $\boldsymbol{z}_k$ and optimized for Poisson negative log-likelihood instead of mean squared error. To obtain predicted firing rates from the smoothed latent states, similar to the $C_z$ readout for behavior decoding, we tested one- and two-layer MLPs, and applied an exponential activation to the output. 3) We train an ensemble of models with different nonlinearity settings and get the final prediction as the mean of the top models from the ensemble based on performance on the validation set (not the test set). Details of model training and hyperparameter selection, including full search ranges, are in Appendix A.12.

We report the results of DPAD with smoothing on three NLB datasets (Pei et al., 2021), along with other baselines in Section 4.5. Similar to our PSID extension, here we focus on the first stage of the method, where dynamics relevant to the secondary signal are extracted (Sani et al., 2024). As such, our optimization involves two steps: (1) learning the bidirectional RNN encoder to extract the latent states $\boldsymbol{x}_k^{smooth}$ and the readout network $C_z$ by optimizing prediction of $\boldsymbol{z}_k$ (2) learning the readout network $C_y$ by optimizing prediction of $\boldsymbol{y}_k$ from the extracted latent states $\boldsymbol{x}_k^{smooth}$.

### 3.6 EVALUATION METRICS

To validate our extensions of PSID in simulations, we generate realizations from 20 random models and evaluate performance using two metrics. First, we confirm that the learned model parameters match the optimal parameters known from ground truth simulated models by computing normalized errors as described in A.14. Second, we confirm that the obtained filtered and smoothed estimates of the secondary signal using the primary signal reach the optimal values that we would obtain from the true model that simulated the data. We use cross-validation and set aside test data distinct from the training data. We then compute the coefficient of determination ($R^2$) between the predicted and true time series of the secondary signal on the test data.

### 3.7 NEURAL DATASETS AND PREPROCESSING

We also evaluate our extensions of PSID and DPAD on several publicly available datasets: Sabes (O'Doherty et al., 2020) and three motor cortex datasets from the NLB (**Area2_Bump**, **MC_Maze**, and **MC_RTT**) (Pei et al., 2021). For the Sabes dataset, we analyzed smoothed spikes, raw local field potential (LFP) and LFP power bands from 7 sessions (details in A.10). Neural features and behavior are z-scored, and decoding performance is evaluated using session-wise cross-validation on held-out test data. For the NLB datasets, we used the version of data with spikes binned at 20 ms and applied a Gaussian smoothing filter to the raw spike counts before feeding them to the model. We apply z-scoring to the Gaussian smoothed spike counts and to behavior, if present.

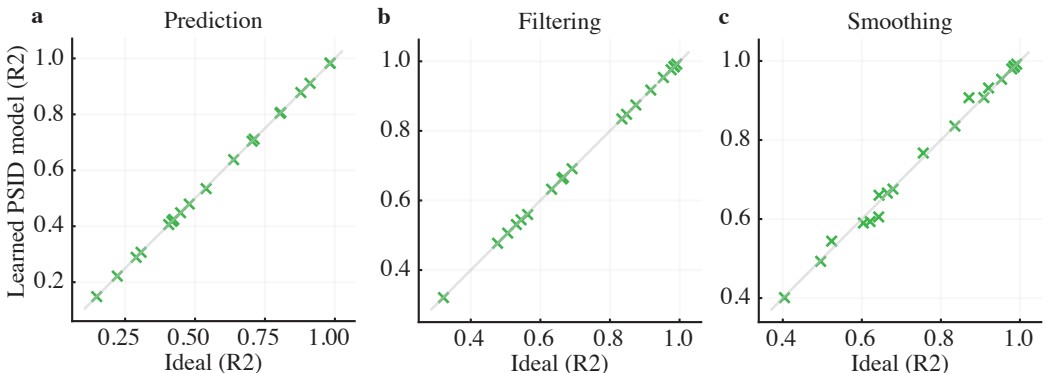

Figure 2: Estimation performance of the secondary signal for (**a**) One-step-ahead prediction ($k|k-1$), (**b**) Filtering ($k|k$), and (**c**) Smoothing ($k|N$), for true models versus the models learned using our extended PSID method. Each point represents one simulated model with random parameters.

## 4 RESULTS

### 4.1 VALIDATION OF PSID WITH FILTERING

As noted in Section 3.6, we simulated 20 random models and performed PSID with filtering. For all parameters of PSID with filtering, including the $C_z K_f$ parameter, as the number of training samples increases, the error converges to smaller and smaller values (Figure 5). Specifically, in this simulation, with a million training samples, the average normalized error for all identifiable parameters converged to below 1%. $K_f$ is an example of a non-identifiable parameter, for which as expected the error does not converge to zero (Figure 5, see Appendix A.14 for more details).

### 4.2 VALIDATION OF FILTERING AND SMOOTHING PSID IN TERMS OF ESTIMATING BEHAVIOR

Next, we used the learned models to compute filtered estimates of the secondary signal $z_k$ and compared them with the true secondary signal in the test set by computing the coefficient of determination ($R^2$) between the two. We repeated this procedure using the true models. As we see in Figure 2b, the filtered estimate of the secondary signal is similar to the performance of the true models. The results are similar to those obtained for the one-step-ahead predictions from PSID (Figure 2a).

Similarly, we used the learned models to compute the smoothed estimate of the secondary signal $z_k$ from the complete time samples of the primary signal. We also did the same using the true models and then computed the $R^2$ between the smoothed estimate of the secondary signal and the true secondary signal in the test set. As we see in Figure 2c, the smoothed estimate of the secondary signal is similar to the performance of the true models, confirming that the learned models also achieve optimal smoothing of the secondary signal.

### 4.3 VALIDATION OF FILTERING AND SMOOTHING PSID IN REAL NEURAL-BEHAVIORAL DATA

After establishing the filtering and smoothing capabilities of PSID on simulated linear systems, we validated the approach on real non-human primate neural-behavioral data from the Sabes dataset (O'Doherty et al., 2020). For each neural modality (spikes, raw LFP, LFP powers), we modeled neural activity and behavior using PSID under prediction, filtering, and smoothing settings.

We compared PSID with linear neural dynamical modeling (NDM), which first fits a linear state-space model to neural activity without considering behavior, and then learns a mapping to behavior. To learn this state-space model, we use the standard subspace identification algorithm (SID) (Van Overschee & De Moor, 1996). Since linear NDM does not explicitly use behavior to learn the neural states, it may less accurately learn the behaviorally relevant neural dynamics that PSID explicitly prioritizes. For both PSID and linear NDM, we used a horizon parameter of $i = 15$ for all analyses.

We first looked at the cross-validated behavior decoding accuracy across state dimensions among powers of 2 up to 16 (Figure 8). Then, we compared the peak accuracy (Figure 3), defined as the decoding accuracy at the latent state dimension where each method reaches within 3% of its maximum decoding accuracy in the training data across all latent state dimensions.

Across all three modalities and both methods, we found that filtering had significantly higher decoding accuracy than one-step-ahead prediction. Additionally, smoothing had significantly higher decoding accuracy than both one-step-ahead prediction and filtering. Moreover, across all modalities, PSID consistently outperformed its linear NDM counterparts, highlighting the importance of PSID's prioritized modeling of behaviorally relevant neural dynamics.

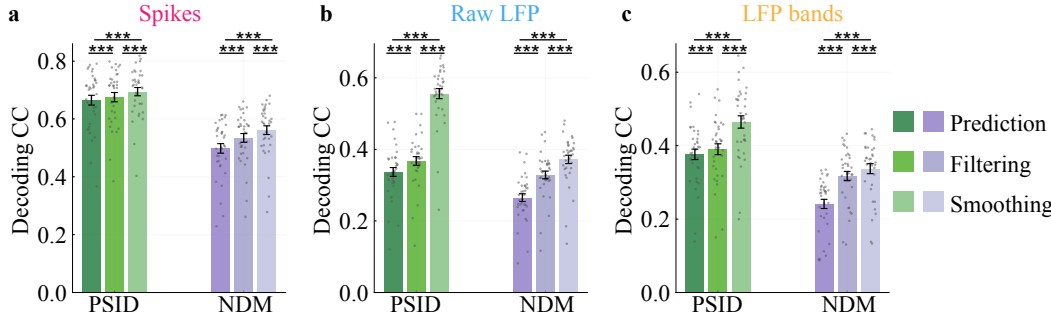

Figure 3: **(a)** Peak cross-validated decoding correlation coefficient (CC) achieved by PSID and linear NDM for prediction, filtering, and smoothing. Bars show the mean and whiskers show the s.e.m. All data points are shown ($N = 35$ session-folds). Asterisks show significance level for a one-sided Wilcoxon signed-rank test (*$P < 0.05$, **$P < 0.005$, ***$P < 0.0005$) The neural modality is spiking activity. **(b)** Same as **(a)** but for raw LFP. **(c)** Same as **(a)** but for LFP power band activity.

### 4.4 EVALUATING DPAD WITH SMOOTHING ON THE SABES DATASET

We analyzed model performance across different latent state dimensions to understand the contributions of nonlinearity and smoothing on the NLB datasets. We compared DPAD, DPAD with smoothing, and nonlinear NDM with Smoothing across all three datasets for both behavior decoding and held-out neural prediction tasks. The nonlinear NDM baseline refers to a model that uses the same nonlinear architecture as DPAD but without prioritization of the secondary signal. Figure 4 shows that DPAD with smoothing, which leverages information from all time points in a trial, consistently outperforms the causal DPAD model across tasks and datasets. In contrast, while nonlinear NDM with smoothing also benefits from smoothing, it lacks prioritized modeling of the secondary signal and is consistently inferior to DPAD with smoothing. These results highlight that smoothing alone is not sufficient—combining smoothing with prioritized modeling, as in DPAD with smoothing, provides the largest performance gains for both behavior decoding and held-out neural prediction.

### 4.5 EVALUATION ON NEURAL LATENTS BENCHMARK

We next compared DPAD with smoothing against other methods on the official NLB leaderboard. In addition to this single-model variant, we also assessed an ensemble version of DPAD with smoothing, which is constructed via validation-based selection (see Appendix A.12 for more details). Table 1 shows results for both behavior decoding measured in $R^2$ and held-out neural prediction measured in co-smoothing bits per spike (co-bps). As baselines, we include LangevinFlow (Song et al., 2025) and Ctrl-TNDM (BAND), which are the current top-performers on the leaderboard, along with several other methods (see Appendix A.11 for a summary). DPAD with smoothing outperformed all methods in behavior decoding in one dataset, and reached competitive near-the-top performances in others. Critically, results listed in Table 1 are all computed by the NLB servers on withheld test splits that no applicant has access to (Pei et al., 2021), which creates a fair comparison and eliminates confounds such as differences in data, preprocessing, metric computation, coding errors, etc.

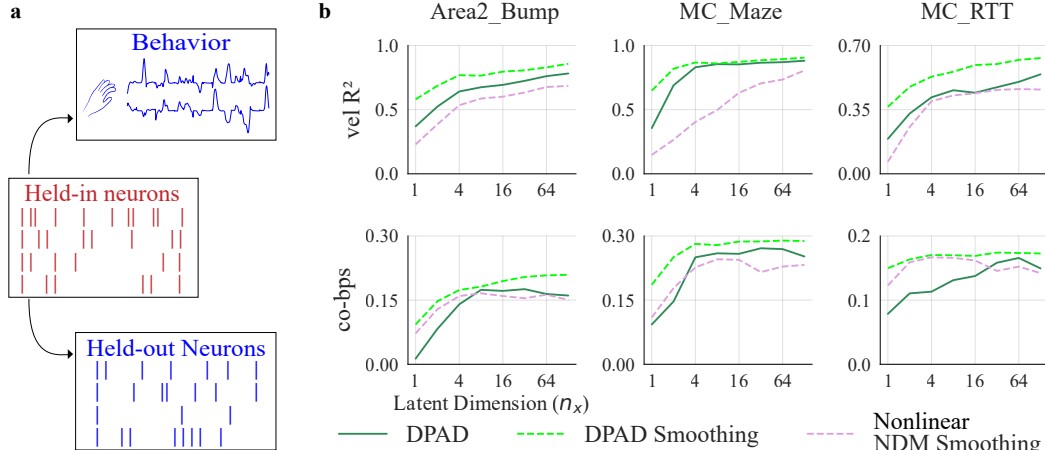

Figure 4: Performance comparison of DPAD with smoothing, DPAD, and nonlinear NDM with smoothing across the datasets and tasks. **(a)** Schematic of the NLB challenge setup: the model is trained on held-in neurons to predict both behavior and held-out neural activity. **(b)** Performance across datasets, with top panels showing behavior decoding ($R^2$) and bottom panels showing held-out neural prediction (co-bps). Smoothing consistently improves performance across tasks, demonstrating the benefit of combining nonlinear modeling with smoothing.

Table 1: Performance on the Neural Latents Benchmark (NLB). **Bold**: best result in the table, Underline: second best result in the table. Asterisk: top result on NLB leaderboard as of Sep 2025. Parentheses: difference of ensemble DPAD with smoothing, with the leaderboard leader.

| Method | Area2_Bump | | MC_Maze | | MC_RTT | |
|---|---|---|---|---|---|---|
| | vel $R^2 \uparrow$ | co-bps↑ | vel $R^2 \uparrow$ | co-bps↑ | vel $R^2 \uparrow$ | co-bps↑ |
| GPFA | 0.6094 | 0.1791 | 0.6613 | 0.2463 | 0.5263 | 0.1769 |
| SLDS | 0.6967 | 0.1816 | 0.7944 | 0.2117 | 0.5365 | 0.1662 |
| AutoLFADS | 0.8565 | 0.2542 | 0.8906 | 0.3554 | 0.6105 | 0.1976 |
| NDT | 0.8623 | 0.2624 | 0.8897 | 0.3597 | 0.6100 | 0.1643 |
| MINT | 0.8803 | 0.2718 | 0.9005 | 0.3295 | 0.6547 | 0.2008 |
| NDT3 | - | - | 0.8065 | 0.2775 | 0.5077 | 0.1695 |
| Ctrl-TNDM (BAND) | **0.8892**[*] | 0.2477 | **0.9252**[*] | 0.3537 | 0.6718[*] | 0.1920 |
| LangevinFlow | 0.8810 | **0.2881**[*] | 0.9000 | **0.3650**[*] | 0.6652 | **0.2010**[*] |
| DPAD Smoothing | 0.8664 | 0.2353 | 0.9230 | 0.3206 | **0.7101** | 0.1784 |
| DPAD Smoothing with Ensemble | 0.8664 (-0.0228) | 0.2586 (-0.0295) | 0.9230 (-0.0022) | 0.3493 (-0.0157) | **0.7101** (+0.0383) | 0.1944 (-0.0066) |

## 5 DISCUSSION

**Summary** We developed analytical extensions of PSID that enable optimal filtering ($\hat{z}_{k|k}$) and optimal smoothing ($\hat{z}_{k|N}$) of a secondary signal from a primary time series, and introduced a nonlinear DPAD variant that supports smoothing. Results in simulations and real neural datasets validated our extended methods, providing a comprehensive framework for linear and nonlinear modeling for all three types of estimation: prediction, filtering, and smoothing.

**Identifiability limitations and open questions** We provided theoretical derivations for our extensions of PSID, including why the two-signal modeling scenario increases identifiability of certain model parameters, leading to optimal filtering/smoothing. However, despite this additional visibility into the internals of the system that is afforded to us by having a secondary time series, we cannot learn all internal parameters uniquely. This is because not all of the internal parameters affect the secondary signal. In cases where they do, we explained how the exact $K_f$ can be identified

(Appendix A.8.4). Identifying the associated $(Q, R, S)$ noise statistics for the stochastic form that give a particular $K_f$ is an interesting follow-up problem that we did not tackle here.

**Shifted PSID baseline vs. optimal filtering** A natural baseline for PSID with filtering is to shift the secondary signal forward by one sample during training and apply the original PSID. As shown in Appendix A.15, this "shifted PSID" improves over ideal one-step-ahead prediction but still falls short of optimal filtering. This is because, in the general case with correlated state and observation noises, the optimal update requires the filter gain $K_f$ and cannot be recovered by time-shifting alone. In our simulations, the proposed PSID with filtering achieves optimal filtering irrespective of noise correlations (Figure 2b).

**Optimality and tracking time-varying dynamics in DPAD** [added in revision] Even given enough training data and model capacity, DPAD can be optimal only if the optimal inference for the system that is being modeled can be written in the form of recursions given in equation 10. One example of this is the optimal Kalman filter for a linear system *at steady-state*. In the special case of linear modeling, DPAD's model (equation 10) reduces to the predictor form representation of the state-space model (equation 4), which is based on a steady-state Kalman filter (equation 20). In this case, it is straightforward to see how DPAD can implement an optimal Kalman filter at steady-state, by directly learning all its steady-state parameters via gradient descent (Appendix A.9). Similarly, the bidirectional RNN in DPAD with smoothing can implement the operations of a Kalman smoother at steady-state (Appendix A.5.1). Finally, with additional latent states, the nonlinear RNNs in DPAD have the capacity to also learn time-varying model dynamics that can be formulated as equation 10, such as a state transition matrix that itself evolves over time with a random walk. DPAD could model such time-varying dynamics by dedicating some latent states to tracking the time-varying system parameters. However, deriving theoretical guarantees on optimality or bounds on errors in modeling such time-varying dynamics and for general nonlinear dynamics are beyond the scope of this work (Appendix A.9).

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

# A APPENDIX

## A.1 BACKGROUND: PREDICTION, FILTERING, AND SMOOTHING

Given a state space model, the problem of *prediction* is finding the optimal estimate of the state $x_k$ at a given time sample $k$ given all past samples of $y$. *Filtering* is the problem of estimating $x_k$ at a given time step $k$ given all samples of $y$ up to and including the current time step $y_k$. Finally, *smoothing* is the problem of estimating $x_k$ at a given time step given samples of $y$ up to a future time step after $k$.

For models with directly measurable states, e.g., kinematics of an object, all three problems have unique solutions. For models with latent states, the exact state is ultimately an internal characteristic of the system and its alternative estimates are only preferable insofar as they can be validated against an observable external characteristic of the system (Katayama, 2006). For example, an estimation of the observation itself $y_k$ based on the estimated latent state gives one measurable way to evaluate the estimated latent state. However, while estimation of $y_k$ using its past values, i.e., *prediction*, is non-trivial, the *filtering* and *smoothing* of $y_k$ is not. Specifically, assuming zero-mean additive observation noises, the best *estimation* of any given sample $y_k$ (in the sense of having minimum expected value of squared error) would simply be its observed value if that sample $y_k$ itself is observed. To confirm this statement, note that the expected squared error of such estimation (i.e., estimating some $x$ as the noisy measured $y = x + \epsilon$) would be the covariance of the additive noise, which is the fundamental minimum error possible. In this one-signal setting thus the filtering and smoothing problems have trivial solutions where the estimated value of $y_k$ is the measured $y_k$ itself.

In contrast, in two-signal identification problems, which is the focus of this work, the objective is to estimate a secondary signal $z_k$ from the primary observations $y_k$. In this setting, filtering and smoothing of $z_k$ are nontrivial and as we show in this work, they can be optimized beyond what an unsupervised one-signal method could achieve.

## A.2 KALMAN FILTER

Given observations $y_0, y_1, \ldots, y_k$, a Kalman filter gives the optimal (in the sense of having the minimum mean squared error) estimate of the latent state $x_{k+1}^s$ as follows (Anderson & Moore, 2012; Åström & Wittenmark, 2013):

$$\hat{x}_{k|k} = \hat{x}_{k|k-1} + K_f(y_k - C_y\hat{x}_{k|k-1}) = (I - K_fC_y)\hat{x}_{k|k-1} + K_fy_k \tag{12a}$$

$$\hat{x}_{k+1|k} = A\hat{x}_{k|k} + \hat{w}_{k|k} = A\hat{x}_{k|k} + K_v(y_k - C_y\hat{x}_{k|k-1}) = (A - KC_y)\hat{x}_{k|k-1} + Ky_k \tag{12b}$$

where Kalman gains $K_f$, $K_v$, and $K$ are defined as

$$K_f \triangleq P_{k|k-1}C_y^T(C_yP_{k|k-1}C_y^T + R)^{-1} \tag{13a}$$

$$K_v \triangleq S(C_yP_{k|k-1}C_y^T + R)^{-1} \tag{13b}$$

$$K \triangleq AK_f + K_v = (AP_{k|k-1}C_y^T + S)(C_yP_{k|k-1}C_y^T + R)^{-1} \tag{13c}$$

and $P_{k|k-1}$ represents the error covariance of the estimated state, defined as:

$$P_{k|k-1} \triangleq \mathbb{E}\left[(\hat{x}_{k|k-1} - x_k^s)(\hat{x}_{k|k-1} - x_k^s)^T\right] \tag{14}$$

This covariance follows the following recursive Riccati equations:

$$P_{k|k} = P_{k|k-1} - P_{k|k-1}C_y^T(C_yP_{k|k-1}C_y^T + R)^{-1}C_yP_{k|k-1} = P_{k|k-1} - K_fC_yP_{k|k-1} \tag{15a}$$

$$P_{k+1|k} = AP_{k|k-1}A^T + Q - (AP_{k|k-1}C_y^T + S)(C_yP_{k|k-1}C_y^T + R)^{-1}(AP_{k|k-1}C_y^T + S)^T$$

$$= AP_{k|k-1}A^T + Q - K(C_yP_{k|k-1}C_y^T + R)K^T. \tag{15b}$$

Initial conditions for the Kalman filter also need to be specified for the above recursive equations to start, but given their limited effect on the steady state performance of stable models, they can usually be chosen as

$$\hat{\boldsymbol{x}}_{0|-1} = \boldsymbol{0}, \qquad P_{0|-1} = I \tag{16}$$

where $\hat{\boldsymbol{x}}_{0|-1}$ is the initial state estimate and $P_{0|-1}$ is the initial error covariance.

For the stationary state space model of equation 1, when the Riccati equations have a stable solution, at steady state, $P_{k+1|k}$ and $P_{k|k}$ converge to steady state values that we denote by $P_p$ and $P$, respectively. The steady state version of equations 15 is thus

$$P = P_p - P_p C_y^T (C_y P_p C_y^T + R)^{-1} C_y P_p \tag{17a}$$

$$P_p = A P_p A^T + Q - (A P_p C_y^T + S)(C_y P_p C_y^T + R)^{-1}(A P_p C_y^T + S)^T . \tag{17b}$$

### A.3 STOCHASTIC VERSUS PREDICTOR FORM FORMULATIONS

Having reviewed the Kalman filter, we can now discuss an important concept. Equations 1-2 are only one of several equivalent ways to formulate the multivariate Gaussian random process $\boldsymbol{y}_k$ as a latent state space model. Specifically, this formulation, which is repeated below, is referred to as the forward stochastic model (Van Overschee & De Moor, 1996):

---

**The stochastic form**

$$\boldsymbol{x}_{k+1}^s = A\boldsymbol{x}_k^s + \boldsymbol{w}_k \tag{18a}$$

$$\boldsymbol{y}_k = C_y \boldsymbol{x}_k^s + \boldsymbol{v}_k \tag{18b}$$

$$\mathbb{E}\left(\begin{bmatrix} \boldsymbol{w}_p \\ \boldsymbol{v}_p \end{bmatrix} \begin{bmatrix} \boldsymbol{w}_q \\ \boldsymbol{v}_q \end{bmatrix}^T\right) = \begin{pmatrix} Q & S \\ S^T & R \end{pmatrix} \delta_{pq} \tag{18c}$$

$$\mathbb{E}[\boldsymbol{x}_k^s(\boldsymbol{x}_k^s)^T] \triangleq \Sigma_x = A\Sigma_x A^T + Q, \tag{19a}$$

$$\mathbb{E}[\boldsymbol{y}_k \boldsymbol{y}_k^T] \triangleq \Sigma_y = C_y \Sigma_x C_y^T + R, \tag{19b}$$

$$\mathbb{E}[\boldsymbol{x}_{k+1}^s \boldsymbol{y}_k^T] \triangleq G_y = A\Sigma_x C_y^T + S. \tag{19c}$$

---

Here, equations 19a-c are obtained by taking covariances and cross covariances from equations 18a-b. These equations specify the relationship between the $Q$, $R$, and $S$ noise covariances with the latent state and observation covariances $\Sigma_x$, $\Sigma_y$, and $G_y$. Specifically, to find $\Sigma_x$, $\Sigma_y$, and $G_y$ based on the former, we can simply use equations 19a-c. Conversely, to find $Q$, $R$, and $S$ based on $\Sigma_x$, $\Sigma_y$, and $G_y$, we can solve the Lyapunov equation (equation 19a) to find a solution for $\Sigma_x$ and then replace that solution in equations 19b-c to find $R$ and $S$, respectively.

An alternative equivalent formulation that describes the exact same second order statistics for $\boldsymbol{y}_k$ is the "forward predictor form" formulation provided below, where the latent state $\boldsymbol{x}_k$ is taken to be the Kalman estimated state, i.e., $\boldsymbol{x}_k \triangleq \hat{\boldsymbol{x}}_{k|k-1}$:

> **The predictor form**
>
> $$\boldsymbol{x}_{k+1} = A\boldsymbol{x}_k + K\boldsymbol{e}_k \qquad (20a)$$
> $$\boldsymbol{y}_k = C_y\boldsymbol{x}_k + \boldsymbol{e}_k \qquad (20b)$$
>
> $$\mathbb{E}[\boldsymbol{x}_k(\boldsymbol{x}_k)^T] \triangleq \tilde{P}_k \qquad (21a)$$
> $$\mathbb{E}[\boldsymbol{e}_k\boldsymbol{e}_k] \triangleq \Sigma_e = \Sigma_y - C_y\tilde{P}_kC_y^T \qquad (21b)$$
>
> $$\tilde{P}_k = A\tilde{P}_{k-1}A^T + (G_y - A\tilde{P}_{k-1}C_y^T)(\Sigma_y - C_y\tilde{P}_{k-1}C_y^T)^{-1}(G_y - A\tilde{P}_{k-1}C_y^T)^T \quad (21c)$$
> $$K_{k-1} = (G_y - A\tilde{P}_{k-1}C_y^T)(\Sigma_y - C_y\tilde{P}_{k-1}C_y^T)^{-1} \qquad (21d)$$

Here, $\boldsymbol{e}_k$ is the part of the observation $\boldsymbol{y}_k$ that is not predictable from past observation samples. $\boldsymbol{e}_k$ is also known as the innovation, which is why this formulation is known as the innovation form. Notably, simply replacing $\boldsymbol{e}_k$ in equation 20a with its definition from equation 20b (i.e., $\boldsymbol{y}_k - C_y\boldsymbol{x}_k$) yields the Kalman filter equation 12b. After that replacement, this formulation is known as the predictor form. Hereafter, for simplicity, we refer to both of these closely related formulations (innovation and predictor forms) as the predictor form.

Equation 21a defines the covariance of the Kalman predicted state (i.e., $\boldsymbol{x}_k$) itself, which is different from the error covariance of the predicted state (i.e., $P_k$). The relation of these two covariances can be derived by taking covariance from the relation between the underlying entities (Van Overschee & De Moor, 1996):

$$\boldsymbol{x}_k^s = \boldsymbol{x}_k + (\boldsymbol{x}_k^s - \boldsymbol{x}_k) \qquad (22a)$$
$$\Sigma_x = \tilde{P}_k + P_k \qquad (22b)$$

where we have used the fact that the Kalman prediction error ($\boldsymbol{x}_k^s - \boldsymbol{x}_k$) is orthogonal to Kalman predicted state ($\boldsymbol{x}_k$). Equation 21b is obtained by taking covariance from equation 20b. Equation 21c is an equivalent formulation of the Riccati equation 15b, related via equation 19. Finally, equation 21d is an alternative equivalent formulation for the Kalman gain equation 13c.

While both the stochastic and predictor forms generate the same second-order statistics for the observations $\boldsymbol{y}_k$, they use different model parameters. It is straightforward to find the predictor form parameters given the stochastic form parameters by simply computing the Kalman filter parameters for the stochastic model (see equations 13-15). This conversion is indeed unique (within a similarity transform; see Appendix A.6) because each model has a specific unique Kalman filter associated with it.

The opposite conversion, from predictor form to stochastic form, is not unique and has an infinite number of solutions, even beyond similarity transforms. This is because the stochastic form is a redundant representation with more parameters than needed to describe the second-order statistics of the observations $\boldsymbol{y}_k$ (Van Overschee & De Moor, 1996; Katayama, 2006). The family of solutions for this conversion is given by Faurre's theorem (Van Overschee & De Moor, 1996).

**Faurre's Theorem:** The set of all state covariance matrices $\Sigma_x$ that generate the same output covariance statistics for $\boldsymbol{y}_k$ is a closed, convex, and bounded set characterized by the inequality:

$$\tilde{P} \leq \Sigma_x \leq \tilde{N}^{-1} \qquad (23)$$

where:

- $\tilde{P}$ is the unique solution to the forward Riccati equation (equation 21c),
- $\tilde{N}$ is the unique solution to the backward Riccati equation (see Van Overschee & De Moor (1996)),
- $\Sigma_x$ is the state covariance matrix for the stochastic form.

For every $\Sigma_x$ satisfying this inequality, the noise covariances for the stochastic form can be constructed by replacing $\Sigma_x$ in equation 19.

Thus, there are infinitely many stochastic models (with different $Q$, $R$, $S$) that generate the same second-order statistics for $\boldsymbol{y}_k$, all parameterized by the choice of $\Sigma_x$ within the bounds above.

The redundancy of the stochastic form in terms of model parameters can also be confirmed by simply counting the number of parameters for stochastic and predictor forms. The stochastic and predictor forms can be summarized with the set of parameters $\{A, C_y, Q, R, S\}$ and $\{A, C_y, K, \Sigma_e\}$, respectively. The $A$ and $C_y$ are shared between them, but the noises are described with $(n_x+n_y)(n_x+n_y+1)/2 = n_x^2/2 + n_x/2 + n_x n_y + n_y^2/2 + n_y/2$ (for $Q, R, S$) versus $n_x n_y + n_y^2/2 + n_y/2$ (for $K, \Sigma_e$) independent parameters (i.e., not counting complex conjugate terms), for stochastic versus predictor forms, respectively. As such, the stochastic form uses $n_x(n_x+1)/2$ more parameters to describe the same $\boldsymbol{y}_k$.

Critically, as far as the time series $\boldsymbol{y}_k$ on its own is concerned, all stochastic representations of the model are equivalent. A key insight presented in this work however is that this is no longer the case in the PSID setting, where a second time series $\boldsymbol{z}_k$ is also measured during modeling. Before discussing that however, we will need to also review Kalman smoothing.

## A.4 BACKWARD STOCHASTIC MODEL

Equivalent to the stochastic form representation of a model (see Appendix A.3), one can also describe the same second order statistics of the observations in terms of a backward stochastic model where the direction of time is reversed (Van Overschee & De Moor, 1996). Specifically, this formulation, which is repeated below, is referred to as the backward stochastic model:

### The backward stochastic form

$$\boldsymbol{x}_{k-1}^b = A^T \boldsymbol{x}_k^b + \boldsymbol{w}_k^b \tag{24a}$$

$$\boldsymbol{y}_k = G_y^T \boldsymbol{x}_k^b + \boldsymbol{v}_k^b \tag{24b}$$

$$\mathbb{E}\left(\begin{bmatrix} \boldsymbol{w}_p^b \\ \boldsymbol{v}_p^b \end{bmatrix}\begin{bmatrix} \boldsymbol{w}_q^b \\ \boldsymbol{v}_q^b \end{bmatrix}^T\right) = \begin{pmatrix} Q^b & S^b \\ (S^b)^T & R^b \end{pmatrix}\delta_{pq} \tag{24c}$$

$$\mathbb{E}[\boldsymbol{x}_k^b(\boldsymbol{x}_k^b)^T] \triangleq (\Sigma_x)^{-1} = A^T(\Sigma_x)^{-1}A + Q^b, \tag{25a}$$

$$\mathbb{E}[\boldsymbol{y}_k\boldsymbol{y}_k^T] \triangleq \Sigma_y = G_y^T(\Sigma_x)^{-1}G_y + R^b, \tag{25b}$$

$$\mathbb{E}[\boldsymbol{x}_{k-1}^b\boldsymbol{y}_k^T] \triangleq (C_y)^T = A^T(\Sigma_x)^{-1}G_y + S^b. \tag{25c}$$

Here, the latent state of the backward model $\boldsymbol{x}_k^b$ is related to that of the forward model, i.e., $\boldsymbol{x}_k^s$, by the following relationship (Van Overschee & De Moor, 1996):

$$\boldsymbol{x}_k^b \triangleq \Sigma_x^{-1}\boldsymbol{x}_k^s \tag{26}$$

Additional derivations for the other relations between the backward and forward stochastic models are provided in Van Overschee & De Moor (1996). What we need to add here is the readout equation for the secondary signal $\boldsymbol{z}_k$ in the backward model. Before we derive this — similar to how the primary readout is derived in Van Overschee & De Moor (1996) — we will recall the readout equation for the secondary signal $\boldsymbol{z}_k$ (equation 1) in the forward stochastic model:

$$\boldsymbol{z}_k = C_z\boldsymbol{x}_k^s + \boldsymbol{\epsilon}_k. \tag{27}$$

We will also need to compute the cross-covariance between the secondary signal $\boldsymbol{z}_k$ and the forward latent state $\boldsymbol{x}_{k+1}^s$, denoted by $G_z$, as follows:

$$G_z \triangleq \mathbb{E}[\boldsymbol{x}_{k+1}^s\boldsymbol{z}_k^T] = \mathbb{E}[(A\boldsymbol{x}_k^s + \boldsymbol{w}_k)(C_z\boldsymbol{x}_k^s + \boldsymbol{\epsilon}_k)^T] \tag{28a}$$

$$= A\Sigma_x C_z^T + S_{xz} \tag{28b}$$

where $S_{xz} \triangleq \mathbb{E}[\boldsymbol{w}_k\boldsymbol{\epsilon}_k^T]$. Finally, we denote the minimum variance estimate of one random variable given the other as $\Pi(.|.)$.

We can then derive the readout equation for the secondary signal $z_k$ in the backward model as follows:

$$z_k = \Pi(z_k \mid x_{k+1}^s) + (z_k - \Pi(z_k \mid x_{k+1}^s)) \tag{29a}$$

$$= \mathbb{E}[z_k(x_{k+1}^s)^T](\mathbb{E}[x_{k+1}^s(x_{k+1}^s)^T])^{-1}x_{k+1}^s + (z_k - \Pi(z_k \mid x_{k+1}^s)) \tag{29b}$$

$$= \mathbb{E}[(C_z x_k^s + \epsilon_k)((x_k^s)^T A^T + w_k^T)]\Sigma_x^{-1}x_{k+1}^s + (z_k - \Pi(z_k \mid x_{k+1}^s)) \tag{29c}$$

$$= (C_z \Sigma_x A^T + S_{xz}^T)\Sigma_x^{-1}x_{k+1}^s + (z_k - \Pi(z_k \mid x_{k+1}^s)) \tag{29d}$$

$$= G_z^T x_k^b + \epsilon_k^b \tag{29e}$$

where $\epsilon_k^b \triangleq z_k - \Pi(z_k \mid x_{k+1}^s)$.

For simplicity in presenting metrics for the learning of the backward model parameters in Appendix A.16, we will denote each parameter of the backward stochastic model as the same symbol as in the forward stochastic model, but in curly braces with a *bw* subscript:

$$\{A\}_{bw} \triangleq A^T, \qquad \{C_y\}_{bw} \triangleq G_y^T, \qquad \{C_z\}_{bw} \triangleq G_z^T, \tag{30a}$$

$$\{G_y\}_{bw} \triangleq C_y^T, \qquad \{\Sigma_y\}_{bw} \triangleq \Sigma_y, \tag{30b}$$

and the Kalman gain parameters are computed per equation 13, but based on the above backwards parameters.

## A.5 KALMAN SMOOTHER

Kalman smoothing provides the optimal estimate of the latent state $x_k^s$ at time $k$ given *all* observations up to the final time $N$, i.e., $\hat{x}_{k|N}$. This is in contrast to the Kalman filter, which provides the optimal estimate of $x_k^s$ at time $k$ given observations up to $k$ ($\hat{x}_{k|k}$), and Kalman prediction, which estimates the next state $x_{k+1}^s$ given observations up to $k$ ($\hat{x}_{k+1|k}$). One widely used formulation for smoothing is the Rauch-Tung-Striebel (RTS) smoother, which we describe below.

**RTS Smoother (Rauch-Tung-Striebel)**  In the RTS smoother (Rauch et al., 1965), after the Kalman filter runs in the forward direction, a second estimation step runs in the reverse time direction on the data to update the Kalman filter state estimations based on all the observed future data. The backward estimation is also recursive and can be formulated as follows:

$$L_k = P_{k|k}A^T(P_{k+1|k})^{-1} \tag{31a}$$

$$P_{k|N} = P_{k|k} + L_k(P_{k+1|N} - P_{k+1|k})L_k^T \tag{31b}$$

$$\hat{x}_{k|N} = \hat{x}_{k|k} + L_k(\hat{x}_{k+1|N} - A\hat{x}_{k|k}) \tag{31c}$$

where $\hat{x}_{k|N}$ and $P_{k|N}$ are the smoothed state estimate and covariance, respectively. Note that the "initial" state of this reverse estimation, i.e., $\hat{x}_{N|N}$, is the last filtered state from the forward pass so it is known when the backwards estimation starts.

A useful interpretation of the RTS smoother is that the smoothed state is a weighted average of the forward (filtered) and backward (smoothed) estimates:

$$\hat{x}_{k|N} = (I - L_k A)\hat{x}_{k|k} + L_k\hat{x}_{k+1|N} \tag{32}$$

Importantly, the backward recursive steps in the RTS formulation (equation 31c) look like a filter, except they are not applied on observed data, $y_k$; rather, they are applied on the Kalman filter states, $\hat{x}_{k|k}$, which are the pseudo-observations of this backward filter. Is it possible to reformulate the Kalman smoothing problem as a forward and backward filtering problem where both filters are applied on the observed data, $y_k$? The answer is yes (Fraser & Potter, 1969).

**Forward-backward (two-filter) smoother**  The same smoothed state estimates as the RTS smoother can also be obtained by combining a forward filter with a backward filter, in an approach called the two-filter or forward-backward smoother (Fraser & Potter, 1969; Kitagawa, 2023). Importantly, the backward filter here is not the same as the backward recursion in the RTS smoother: it uses different state and covariance variables, which we denote with a superscript $b$.

**Backward filter** As in the RTS formulation, the forward filter is a Kalman filter. The backward filter, proceeds from $N$ to 1 and is defined as follows:

INITIAL CONDITION:

$$\hat{\boldsymbol{x}}_{N|N+1}^b = \boldsymbol{0}, \qquad P_{N|N+1}^b = 0 \tag{33}$$

UPDATE STEP:

$$\hat{\boldsymbol{x}}_{k|k}^b = \hat{\boldsymbol{x}}_{k|k+1}^b + C_y^T R^{-1} \boldsymbol{y}_k \tag{34a}$$

$$P_{k|k}^b = P_{k|k+1}^b + C_y^T R^{-1} C_y \tag{34b}$$

PREDICTION STEP:

$$J_k = P_{k|k}^b (P_{k|k}^b + Q^{-1})^{-1} \tag{35a}$$

$$\hat{\boldsymbol{x}}_{k-1|k}^b = A^T (I - J_k) \hat{\boldsymbol{x}}_{k|k}^b \tag{35b}$$

$$P_{k-1|k}^b = A^T (I - J_k) P_{k|k}^b A \tag{35c}$$

Note that this formulation from (Kitagawa, 2023) assumes that there is no cross-correlation between the state and observation noises (i.e., $S = 0$).

FORWARD-BACKWARD WEIGHTED AVERAGE SMOOTHER After running both the forward (Kalman) and backward filters, the smoothed state and covariance at each time $k$ can be computed as a weighted average:

$$P_{k|N} = \left( P_{k|k}^{-1} + P_{k|k+1}^b \right)^{-1} \tag{36a}$$

$$\hat{\boldsymbol{x}}_{k|N} = P_{k|N} P_{k|k}^{-1} \hat{\boldsymbol{x}}_{k|k} + P_{k|N} \hat{\boldsymbol{x}}_{k|k+1}^b \tag{36b}$$

where $\hat{\boldsymbol{x}}_{k|k+1}^b$ and $P_{k|k+1}^b$ are the backward filter state and covariance, and $\hat{\boldsymbol{x}}_{k|k}$ and $P_{k|k}$ are the forward (Kalman) filter state and covariance. Note that the backward filter is distinct from the RTS smoother variables, but yield the same optimal smoothed estimate $\hat{\boldsymbol{x}}_{k|N}$.

Also note that although we use the notation $P_{k|k+1}^b$, this quantity is not the covariance of any quantity, rather it is the *inverse* covariance of the backward filter, which is why it is initialized with $\boldsymbol{0}$ in equation 33. This alternative formulation for a Kalman filter that is based on inverse covariances is known as the *information filter* (Anderson & Moore, 2012). Nevertheless, unlike in the RTS formulation, in the two-filter formulation, the backward pass is applied to the observations, just like the forward pass. Finally, it is also worth noting that the backward filter in the forward-backward smoother formulation is different from the filter associated with the backward stochastic model (Van Overschee & De Moor, 1996) that is equivalent to equation 1 (i.e., the backward system is a different model).

The forward-backward smoother formulation is notable because it is closely related to the method we develop in this work. Briefly, in the forward-backward smoother literature, the backward filter parameters are based on the state-space model parameters (as shown in equations 34-35). In contrast, in this work, we learn both the forward and backward filter parameters from the data.

### A.5.1 STEADY-STATE TWO-FILTER KALMAN SMOOTHER [ADDED IN REVISION]

It is helpful for our discussion in Appendix A.8.6 to establish the steady-state behavior, i.e., not close to the beginning or end of the data, for the Kalman smoother. We will do so based on the two-filter formulation in equations 33-36.

For the forward filter, which is the standard Kalman filter, the steady-state behavior is given in equation 17.

To describe the backward filter, we next find the steady-state values for $P_{k|k+1}^b$ and $P_{k|k}^b$, which we denote as $P_p^b$ and $P^b$, respectively. Substituting the definitions of terms from other equations into

equation 35c, we get

$$P_{k-1|k-1}^b = P_{k-1|k}^b + C_y^T R^{-1} C_y \tag{37}$$

$$P_{k-1|k-1}^b = A^T(I - J_k)P_{k|k}^b A + C_y^T R^{-1} C_y \tag{38}$$

$$P_{k-1|k-1}^b = A^T(I - P_{k|k}^b(P_{k|k}^b + Q^{-1})^{-1})P_{k|k}^b A + C_y^T R^{-1} C_y. \tag{39}$$

Substituting $P_{k-1|k-1}^b$ and $P_{k|k}^b$ with $P^b$, we get

$$P^b = A^T(I - P^b(P^b + Q^{-1})^{-1})P^b A + C_y^T R^{-1} C_y \tag{40}$$

$$= A^T P^b A - A^T P^b(P^b + Q^{-1})^{-1}P^b A + C_y^T R^{-1} C_y. \tag{41}$$

Equation 40 is a discrete-time Riccati equation, which has a solution under the usual conditions. Based on equation 35c, we can also write the steady-state expression for $P_p^b$ in terms of $P^b$ as

$$P_p^b = A^T(I - J_k)P^b A \tag{42}$$

$$= A^T(I - P^b(P^b + Q^{-1})^{-1})P^b A. \tag{43}$$

The above gives the steady-state (far from the beginning) behavior for the forward and backward filters in the two-filter Kalman smoother. Replacing these values into equation 36 gives

$$P_s = \left(P^{-1} + P_p^b\right)^{-1} \tag{44a}$$

$$\hat{x}_{k|N} = P_s P^{-1} \hat{x}_{k|k} + P_s \hat{x}_{k|k+1}^b. \tag{44b}$$

This means that after convergence in each direction, i.e., when we are not very close to the beginning or end of the data, the optimal Kalman smoothed state is a weighted average, *with a fixed weight*, of the states of a forward and a backward Kalman filter. In Appendix A.8.6, we explain how our PSID with smoothing training leverages this steady-state behavior for learning.

### A.6 SIMILARITY TRANSFORMS AND EQUIVALENT MODELS BEYOND THEM

Latent state space models are a fundamentally redundant representation, in the sense that one could write infinitely many different state space equations like equation 1 that have different parameters but are equivalent and describe the exact same second order statistics for observation time series $y_k$ and $z_k$ (Van Overschee & De Moor, 1996; Katayama, 2006).

Since the latent state $x_k$ is by definition not measured and does not correspond to any physical quantity all latent state space models that describe the statistics of the observed data (e.g., $y_k$) are equally valid, regardless of their exact latent state. In the one-signal setting, only $y_k$ is observed and thus all models that describe the same second-order statistics of $y_k$ (per Faurre's theorem) are equally valid. In the two-signal setting of PSID, only models are equally valid that further produce the same cross-correlative statistics for the two signals, which would mean that they yield similar conditional probability for $z_k$ given $y_{1:N}$. As we show in this work, this allows us to narrow down the parameter space and find models that are optimal in the prediction of $z_k$ using $y_k$.

### A.7 SYSTEM IDENTIFICATION AND INTERNAL VERSUS EXTERNAL CHARACTERISTICS OF THE MODEL

System identification or model fitting is the problem of finding a set of model parameter that represent a given training data well. As explained in Appendix A.3, certain model representations are more redundant than others, meaning that there are more ways to describe the same data statistics using them. Specifically, the stochastic form latent state space model (equation 18) is a redundant representation, with infinitely many sets of $\{Q, R, S\}$ parameters giving the same second order statistics of $y_k$ (see Faurre's theorem). For this reason, the $\{Q, R, S\}$ parameters are not uniquely identifiable regardless of the method used for learning the model and the available training data. In other words, these parameters are internal characteristics of the stochastic form model and thus do not have a one-to-one manifestation on any measurable property of the system (Katayama, 2006). In

contrast, the Kalman filter that is optimal for any stable Gaussian random process $\boldsymbol{y}_k$ can be uniquely estimated, which is why the predictor form parameters are all external characteristics of the system and are thus uniquely identifiable (within a similarity transform).

Examples of uniquely identifiable (within a similarity transform) model parameters include, $\Sigma_y$, $\Sigma_e$, $K$, $C_y$, and $A$. Notably, unlike the total Kalman gain $K$, its components $K_f$ and $K_v$ (equation 13c) are *not* uniquely identifiable. This has an important ramification for Kalman prediction versus Kalman filtering. While the Kalman prediction (12b) only relies on uniquely identifiable parameters (i.e., $\{A, C_y, K\}$), the update step needed for Kalman filtering (12a) relies on $K_f$, which is not uniquely identifiable. This means that given time series $\boldsymbol{y}_k$ from a system with latent states, the optimal Kalman filter is uniquely identifiable, whereas there are infinitely many Kalman filters associated with that unique Kalman predictor that are equivalent in terms of how they describe $\boldsymbol{y}_k$. This is also intuitively clear, because given a sample of the time series $\boldsymbol{y}_k$, estimating that same time step given its true observed value is a trivial problem that does not require a filter: the optimal estimation of the denoised value of $\boldsymbol{y}_k$ (i.e., $C_y\boldsymbol{x}_k$) given $\boldsymbol{y}_k$ (i.e., $C_y\boldsymbol{x}_k + \boldsymbol{v}_k$) is simply $\boldsymbol{y}_k$ itself, which would yield the minimum possible expected error of $\boldsymbol{v}_k$. This is because $\boldsymbol{v}_k$ is white, and thus no amount of additional observations from other samples besides $\boldsymbol{y}_k$ can provide any information about $\boldsymbol{v}_k$, making it the minimum possible error. The same holds for optimal smoothing for $\boldsymbol{y}_k$ given $\boldsymbol{y}_k$ itself. We emphasize that this triviality of filtering/smoothing and the un-identifiability of an optimal filter/smoother is only the case for systems with *latent* states, not those with measurable states.

The above is only true when only one time series is available. In the PSID setting, where a second time series $\boldsymbol{z}_k$ is available, the joint second order statistics of the two time series are the objective of identification, and this expanded scope disambiguates the identification problem compared with the one-signal cases. In the PSID setting, we further assume that the secondary signal is only measured during training, and only the primary signal is measured during inference. In this setting, non-trivial filtering and smoothing problems can be defined as follows: the optimal filtering is the best estimate of $\boldsymbol{z}_k$ given all samples of $\boldsymbol{y}_k$ up to $k$. The optimal smoothing is the best estimate of $\boldsymbol{z}_k$ given all samples of $\boldsymbol{y}_k$ up to $N$. In other words, in the PSID setting, the presence of a secondary time series $\boldsymbol{z}_k$ during system identification creates a non-trivial filtering and smoothing problem for that secondary time series. As we will show here, this means that otherwise unidentifiable parameters such as $K_f$ become partially identifiable (to the extent that they are related to $\boldsymbol{z}_k$).

## A.8 PSID

Preferential Subspace Identification (PSID) is a system identification method designed to model the dynamics of two time series, $\boldsymbol{y}_k \in \mathbb{R}^{n_y}$ and $\boldsymbol{z}_k \in \mathbb{R}^{n_z}$, the latter of which is not expected to be measured during inference. A key use-case for PSID is modeling neural-behavioral data for use in brain-machine interfaces, where the behavior signal is often a target for decoding and is not measured during inference. However, the method is general and can be applied to any pair of time series.

The key insight of PSID is to identify the dynamics of the primary signal $\boldsymbol{y}_k$ while dissociating dynamics that are relevant to the secondary signal $\boldsymbol{z}_k$ from those that are unrelated to $\boldsymbol{z}_k$. PSID further prioritizes learning the dynamics that are shared between the two time series.

PSID operates in two stages. In the first stage, dynamics that are shared between the two time series are extracted via a projection of future behavior $\boldsymbol{z}_k$ onto corresponding past neural activity $\boldsymbol{y}_k$. In the second stage, any residual dynamics in neural activity that are not explained by the latent states extracted in the first stage are explained using additional latent states. The second stage identifies these additional states by projecting future residual activity onto past neural activity.

In the first stage, a pre-specified number of latent states, denoted by $n_1$, are extracted. In the second stage, an additional pre-specified number of latent states, denoted by $n_x - n_1$, are extracted. After all parameters are learned, the overall model takes the form of equation 45, which is equivalent to equation 1.

$$
\begin{cases}
\begin{bmatrix} \hat{\boldsymbol{x}}_{k+1}^{(1)} \\ \hat{\boldsymbol{x}}_{k+1}^{(2)} \end{bmatrix} &= \begin{bmatrix} A_{11} & A_{12} \\ A_{21} & A_{22} \end{bmatrix} \begin{bmatrix} \hat{\boldsymbol{x}}_{k}^{(1)} \\ \hat{\boldsymbol{x}}_{k}^{(2)} \end{bmatrix} + \begin{bmatrix} \boldsymbol{w}_{k}^{(1)} \\ \boldsymbol{w}_{k}^{(2)} \end{bmatrix} \\[2ex]
\boldsymbol{y}_k &= \begin{bmatrix} C_y^{(1)} & C_y^{(2)} \end{bmatrix} \begin{bmatrix} \hat{\boldsymbol{x}}_{k}^{(1)} \\ \hat{\boldsymbol{x}}_{k}^{(2)} \end{bmatrix} + \boldsymbol{v}_k \\[2ex]
\boldsymbol{z}_k &= \begin{bmatrix} C_z^{(1)} & C_z^{(2)} \end{bmatrix} \begin{bmatrix} \hat{\boldsymbol{x}}_{k}^{(1)} \\ \hat{\boldsymbol{x}}_{k}^{(2)} \end{bmatrix} + \boldsymbol{\epsilon}_k
\end{cases}
\tag{45}
$$

Once the PSID model parameters are learned, at inference time, a Kalman filter per equation 12b can be used to extract the latent states from the neural activity and in turn predict behavior from the latent states. The latent states can simply be multiplied by the parameter $C_z$ to obtain behavior predictions:

$$
\hat{\boldsymbol{z}}_k = C_z \hat{\boldsymbol{x}}_k. \tag{46}
$$

### A.8.1 ONE-STEP-AHEAD PREDICTION VERSUS FILTERING VERSUS SMOOTHING IN THE PSID SETTING

As described in Appendix A.7, not all model parameters are uniquely identifiable, because for some of them, there are infinitely many equivalent solutions. In the context of prediction, similar to one-signal system identification, all PSID parameters are uniquely identifiable. However, in the context of filtering in the PSID setting, there is a fundamental difference compared to the normal one-signal system identification. The difference is that, given the existence of the second time series $\boldsymbol{z}_k$, and the fact that in the first stage of PSID we are optimizing for dynamics of $\boldsymbol{y}_k$ that are relevant to that second time series, there is now a meaningful distinction between all the equivalent models (unlike in Appendix A.3).

These alternative models correspond to different stochastic form models, each with their own Kalman filter and predictor. While the Kalman predictor parameter $K$ is uniquely identifiable even in the one-signal setting, the Kalman filter parameter $K_f$ is not uniquely identifiable (Appendix A.7). In other words, all alternative stochastic form models have the same $K$ (within a similarity transform), while they do not have the same $K_f$. Moreover, these models are not all identical in terms of behavior prediction. Unlike the one-signal setting, in the PSID setting, we can identify the model(s) that are most predictive of behavior because (during training) we have access to the secondary time series $\boldsymbol{z}_k$. Indeed, the objective of the PSID algorithm is to optimize the prediction of this secondary time series. In the case of filtering and smoothing, the objective of the extended PSID algorithm we develop in this work is to estimate the secondary signal using the primary signal samples up to the same sample (filtering) or up to the final sample number $N$ (smoothing).

### A.8.2 THE MORE GENERAL SOLUTION FOR $C_z K_f$ IN PSID WITH FILTERING

Here, we will first review the simpler approach explained in Section 3.4, before explaining a more general version. As noted in Section 3.4, the optimization that we want to solve is that of equation 5. Replacing $\hat{\boldsymbol{x}}_{k|k}$ from equation 12a gives:

$$
\begin{aligned}
& \arg\min_{K_f} \sum_k \| \boldsymbol{z}_k - C_z \hat{\boldsymbol{x}}_{k|k} \|_2^2 \\
=\ & \arg\min_{K_f} \sum_k \| \boldsymbol{z}_k - C_z (\hat{\boldsymbol{x}}_{k|k-1} + K_f(\boldsymbol{y}_k - C_y \hat{\boldsymbol{x}}_{k|k-1})) \|_2^2 \\
=\ & \arg\min_{K_f} \sum_k \| \boldsymbol{z}_k - \hat{\boldsymbol{z}}_{k|k-1} - C_z K_f(\boldsymbol{y}_k - \hat{\boldsymbol{y}}_{k|k-1}) \|_2^2 \\
=\ & \arg\min_{K_f} \sum_k \| \tilde{\boldsymbol{z}}_{k|k-1} - C_z K_f \tilde{\boldsymbol{y}}_{k|k-1} \|_2^2.
\end{aligned}
\tag{47}
$$

As explained in Section 3.4, learning $C_z K_f$ directly is sufficient as far as optimal filtered (i.e., $k|k$) estimation of $\boldsymbol{z}_k$ from $\boldsymbol{y}_k$ is of interest. As such, we can solve the optimization in equation 47 for $C_z K_f$ instead of $K_f$. Indeed, the linear minimum mean squared error estimate for this optimization has a closed-form solution that gives us the optimal $C_z K_f$ as follows:

$$
C_z K_f = \arg\min_M \| \tilde{Z} - M\tilde{Y} \|_F^2 = \tilde{Z}\tilde{Y}^T (\tilde{Y}\tilde{Y}^T)^{-1} \tag{48}
$$

where $\tilde{Z}$ and $\tilde{Y}$ are wide matrices, the columns of which consist of $\tilde{\boldsymbol{z}}_{k|k-1}$ and $\tilde{\boldsymbol{y}}_{k|k-1}$, respectively, for all training samples.

One critical point is that the linear minimum mean squared estimate noted in the equation above may not be correct if $n_y > n_x$ and $n_z > n_x$, in the sense that it may have a rank larger than $n_x$, whereas we expect the rank of $C_z K_f$ to be at most equal to $n_x$, or more precisely at most $min(n_x, n_y, n_z)$. Therefore, as noted in Section 3.4, instead of using the linear minimum mean squared estimate, we use a reduced-rank regression (RRR) solution to enforce the rank of $C_z K_f$ to be at most $n_x$ (Figure 1a).

Finally, we can use a more general version of equation 47 where we learn $\Gamma_z K_f$, where $\Gamma_z$ is the extended observability matrix for the pair $(C_z, A)$, instead of learning $C_z K_f$. This general version can be formulated as follows:

$$
\begin{aligned}
& \arg\min_{K_f} \sum_k \sum_{l=0}^{i-1} \| \boldsymbol{z}_{k+l} - C_z A^l \hat{\boldsymbol{x}}_{k|k} \|_2^2 \\
=\ & \arg\min_{K_f} \sum_k \sum_{l=0}^{i-1} \| \boldsymbol{z}_{k+l} - C_z (A^l \hat{\boldsymbol{x}}_{k|k-1} + A^l K_f(\boldsymbol{y}_k - C_y \hat{\boldsymbol{x}}_{k|k-1})) \|_2^2 \\
=\ & \arg\min_{K_f} \sum_k \sum_{l=0}^{i-1} \| \boldsymbol{z}_{k+l} - \hat{\boldsymbol{z}}_{k+l|k-1} - C_z A^l K_f(\boldsymbol{y}_k - \hat{\boldsymbol{y}}_{k|k-1}) \|_2^2 \\
=\ & \arg\min_{K_f} \sum_k \sum_{l=0}^{i-1} \| \tilde{\boldsymbol{z}}_{k+l|k-1} - C_z A^l K_f \tilde{\boldsymbol{y}}_{k|k-1} \|_2^2 \\
=\ & \arg\min_{K_f} \sum_k \| \underbrace{\begin{bmatrix} \tilde{\boldsymbol{z}}_{k|k-1} \\ \tilde{\boldsymbol{z}}_{k+1|k-1} \\ \vdots \\ \tilde{\boldsymbol{z}}_{k+i-1|k-1} \end{bmatrix}}_{\triangleq \tilde{\boldsymbol{z}}_{k|k-1,i}} - \Gamma_z K_f \tilde{\boldsymbol{y}}_{k|k-1} \|_2^2.
\end{aligned}
\tag{49}
$$

Here, $i$ is the PSID hyperparameter called the horizon (Sani et al., 2021), and $\Gamma_z$, i.e., the extended observability matrix for the pair $(C_z, A)$, is defined as:

$$
\Gamma_z = \begin{bmatrix} C_z \\ C_z A \\ \vdots \\ C_z A^{i-1} \end{bmatrix}.
\tag{50}
$$

This more general optimization can be converted to matrix form as

$$
\Gamma_z K_f = \arg\min_M \| \tilde{Z}_{1:i} - M\tilde{Y} \|_F^2 = \tilde{Z}_{1:i}\tilde{Y}^T (\tilde{Y}\tilde{Y}^T)^{-1}
\tag{51}
$$

where $\tilde{Z}_{1:i}$ and $\tilde{Y}$ are wide matrices, with $\tilde{Y}$ defined as in equation 48 and $\tilde{Z}_{1:i}$ defined similarly with its columns consisting of the tall vector $\tilde{\boldsymbol{z}}_{k|k-1,i}$, for all training samples. In other words, block row number $l+1$ in $\tilde{Z}_{1:i}$, which we denote as $\tilde{Z}_{l:l+1}$, consists of samples of $\tilde{\boldsymbol{z}}_{k+l|k-1}$ in its columns. Note that here, both $\tilde{Z}_{1:i}$ and $\tilde{Y}$ would have $N-i$ columns, since we can't form $\tilde{\boldsymbol{z}}_{k|k-1,i}$ for the last $i$ samples of the training data.

Rather than using the unconstrained solution given in equation 51 for illustration, we solve for $\Gamma_z K_f$ using RRR as before to enforce the rank constraint. We then take the first $n_z$ rows of the learned $\Gamma_z K_f$ as the final learned $C_z K_f$. This more general approach has the benefit that with a large enough $i$, the rank of $\Gamma_z K_f$ is not limited by $n_z$, but rather can be as large as $min(n_x, n_y)$, accommodating the full rank of $K_f$, the identification of which we will discuss next in Appendix A.8.4.

### A.8.3   FORMAL PROOFS FOR PSID WITH FILTERING [ADDED IN REVISION]

In our results (Figure 2b), we empirically validated that PSID with filtering reaches the performance of ideal filtering (i.e., with true model parameters). Here, we provide a formal proof that the filtered estimates given in PSID with filtering are optimal in the sense of being the *linear* minimum variance unbiased estimates of the secondary signal $\boldsymbol{z}_k$, given the primary signal observations $\boldsymbol{y}_0, \boldsymbol{y}_1, \ldots, \boldsymbol{y}_k$. Moreover, since for jointly Gaussian variables the *linear* minimum variance estimate coincides with the minimum variance estimate (Anderson & Moore, 2012), if we assume that $\boldsymbol{\epsilon}_k$ and thus $\boldsymbol{z}_k$ are Gaussian, then our estimates will also be the overall minimum variance estimates. However, we will not make that assumption to keep our definition of $\boldsymbol{\epsilon}_k$ general, as in (Sani et al., 2021).

In our proofs, we will use the following lemmas:

**Lemma 1.** *For random vectors $\boldsymbol{a}$ and $\boldsymbol{b}$, with the mean and covariance*

$$\mathbb{E}\left\{\begin{bmatrix} \boldsymbol{a} \\ \boldsymbol{b} \end{bmatrix}\right\} \triangleq \begin{bmatrix} \boldsymbol{\mu}_a \\ \boldsymbol{\mu}_b \end{bmatrix}, \quad \mathbb{E}\left\{(\begin{bmatrix} \boldsymbol{a} \\ \boldsymbol{b} \end{bmatrix} - \begin{bmatrix} \boldsymbol{\mu}_a \\ \boldsymbol{\mu}_b \end{bmatrix})(\begin{bmatrix} \boldsymbol{a} \\ \boldsymbol{b} \end{bmatrix} - \begin{bmatrix} \boldsymbol{\mu}_a \\ \boldsymbol{\mu}_b \end{bmatrix})^T\right\} \triangleq \begin{bmatrix} \Sigma_a & \Sigma_{ab} \\ \Sigma_{ba} & \Sigma_b \end{bmatrix}, \quad (52)$$

*the linear minimum variance unbiased estimate of $\boldsymbol{b}$ given $\boldsymbol{a}$ is:*

$$\mathbb{E}^*\{\boldsymbol{b}|\boldsymbol{a}\} = \boldsymbol{\mu}_b + \Sigma_{ba}\Sigma_{aa}^{-1}(\boldsymbol{a} - \boldsymbol{\mu}_a). \quad (53)$$

If $\Sigma_{aa}$ is not invertible, $\Sigma_{ba}\Sigma_{aa}^{-1}$ is replaced with $\Sigma_{ba}\Sigma_{aa}^{\dagger} + \bar{\Sigma}$, for any $\bar{\Sigma}$ with $\bar{\Sigma}\Sigma_{aa} = \mathbf{0}$.

Lemma 1 is proved in Anderson & Moore (2012) as *Theorems 2.1 and 2.3 of Chapter 5*. The fact that the minimum variance unbiased estimator is the conditional mean is proved as *Theorem 3.1 of Chapter 2* in Anderson & Moore (2012).

**Lemma 2.** *Given the state-space model of equation 1 and the associated Kalman filter equations 12-17, we have:*

$$\mathbb{E}\{\boldsymbol{x}_k^s \mid \boldsymbol{y}_0 \ldots \boldsymbol{y}_{k-1}\} = \mathbb{E}\{\boldsymbol{x}_k^s \mid \hat{\boldsymbol{x}}_{k|k-1}\} = \hat{\boldsymbol{x}}_{k|k-1}. \quad (54)$$

*Proof.* The proof is evident based on the Kalman filter recursions. The expected value on the left hand side is equal to $\hat{\boldsymbol{x}}_{k|k-1}$, as defined by the Kalman filter recursions of equations 12. Replacing $\boldsymbol{x}_k^s = \hat{\boldsymbol{x}}_{k|k-1} + \tilde{\boldsymbol{x}}_{k|k-1}$ in the right hand side and noticing that the Kalman filter state estimation error $\tilde{\boldsymbol{x}}_{k|k-1}$ is zero mean and orthogonal to the state estimation $\hat{\boldsymbol{x}}_{k|k-1}$ gives $\mathbb{E}\{\boldsymbol{x}_k^s \mid \hat{\boldsymbol{x}}_{k|k-1}\} = \mathbb{E}\{\hat{\boldsymbol{x}}_{k|k-1} \mid \hat{\boldsymbol{x}}_{k|k-1}\} = \hat{\boldsymbol{x}}_{k|k-1}$. This completes the proof. $\qquad \square$

Equipped with these Lemmas, we will first prove the relationship for the optimal one-step-ahead prediction of $\boldsymbol{z}_k$ given past observations $\boldsymbol{y}_0, \ldots, \boldsymbol{y}_{k-1}$:

**Theorem 1.** *Given the state-space model of equation 1 and the associated Kalman filter equations 12-17, the linear minimum variance estimate of $\boldsymbol{z}_k$ given past observations $\boldsymbol{y}_0, \ldots, \boldsymbol{y}_{k-1}$ is given by*

$$\hat{\boldsymbol{z}}_{k|k-1} \triangleq \mathbb{E}^*\{\boldsymbol{z}_k \mid \boldsymbol{y}_0, \ldots, \boldsymbol{y}_{k-1}\} = \mathbb{E}^*\{\boldsymbol{z}_k \mid \hat{\boldsymbol{x}}_{k|k-1}\} = C_z\hat{\boldsymbol{x}}_{k|k-1}. \quad (55)$$

*Proof.* Replacing $\boldsymbol{z}_k = C_z\boldsymbol{x}_k + \boldsymbol{\epsilon}_k$ gives

$$\mathbb{E}^*\{\boldsymbol{z}_k \mid \boldsymbol{y}_0, \ldots, \boldsymbol{y}_{k-1}\} = C_z\mathbb{E}^*\{\boldsymbol{x}_k^s \mid \boldsymbol{y}_0, \ldots, \boldsymbol{y}_{k-1}\} + \mathbb{E}^*\{\boldsymbol{\epsilon}_k \mid \boldsymbol{y}_0, \ldots, \boldsymbol{y}_{k-1}\} \quad (56a)$$

$$= C_z\mathbb{E}^*\{\boldsymbol{x}_k^s \mid \hat{\boldsymbol{x}}_{k|k-1}\} + \mathbb{E}^*\{\boldsymbol{\epsilon}_k \mid \boldsymbol{y}_0, \ldots, \boldsymbol{y}_{k-1}\}, \quad (56b)$$

where we have used Lemma 2 to get the second line. Further, $\boldsymbol{\epsilon}_k$ is independent of all past observations and noises, thus its expected value is the same when conditioned on $\hat{\boldsymbol{x}}_{k|k-1}$ instead of $\boldsymbol{y}_0, \ldots, \boldsymbol{y}_{k-1}$, giving

$$\mathbb{E}^*\{\boldsymbol{z}_k \mid \boldsymbol{y}_0, \ldots, \boldsymbol{y}_{k-1}\} = C_z\mathbb{E}^*\{\boldsymbol{x}_k^s \mid \hat{\boldsymbol{x}}_{k|k-1}\} + \mathbb{E}^*\{\boldsymbol{\epsilon}_k \mid \hat{\boldsymbol{x}}_{k|k-1}\} \quad (57a)$$

$$= \mathbb{E}^*\{C_z\boldsymbol{x}_k^s + \boldsymbol{\epsilon}_k \mid \hat{\boldsymbol{x}}_{k|k-1}\} = \mathbb{E}^*\{\boldsymbol{z}_k \mid \hat{\boldsymbol{x}}_{k|k-1}\}. \quad (57b)$$

This proves the first equality in the theorem. Instead, using Lemma 2 on the first line of equation 57a, and replacing $\mathbb{E}^*\{\boldsymbol{\epsilon}_k \mid \hat{\boldsymbol{x}}_{k|k-1}\} = \mathbf{0}$, we get the second equality, completing the proof. $\qquad \square$

Equipped with Theorem 1, we will next prove the relation for the optimal filtered estimation of $\boldsymbol{z}_k$ (i.e., $\hat{\boldsymbol{z}}_{k|k}$):

**Theorem 2.** *Given the state-space model of equation 1 and the associated Kalman filter equations 12-17, the linear minimum variance unbiased prediction of $\boldsymbol{z}_k$ given $\boldsymbol{y}_0 \ldots \boldsymbol{y}_k$ is given by*

$$\hat{\boldsymbol{z}}_{k|k} = \hat{\boldsymbol{z}}_{k|k-1} + \mathbb{E}\{\tilde{\boldsymbol{z}}_{k|k-1}\tilde{\boldsymbol{y}}_{k|k-1}^T\}\mathbb{E}\{\tilde{\boldsymbol{y}}_{k|k-1}\tilde{\boldsymbol{y}}_{k|k-1}^T\}^{-1}\tilde{\boldsymbol{y}}_{k|k-1}. \quad (58)$$

*Proof.* Again, based on Lemma 1, the linear minimum variance unbiased estimate of $\boldsymbol{z}_k$ given $\boldsymbol{y}_0 \ldots \boldsymbol{y}_k$ is given by

$$\hat{\boldsymbol{z}}_{k|k} = \mathbb{E}^*\{\boldsymbol{z}_k \mid \boldsymbol{y}_0 \ldots \boldsymbol{y}_{k-1}, \boldsymbol{y}_k\}. \quad (59)$$

Writing $\boldsymbol{y}_k$ as $C_y\hat{\boldsymbol{x}}_{k|k-1} + \tilde{\boldsymbol{y}}_{k|k-1}$ and using Lemma 2, we get

$$\hat{\boldsymbol{z}}_{k|k} = \mathbb{E}^*\{\boldsymbol{z}_k \mid \hat{\boldsymbol{x}}_{k|k-1}, \tilde{\boldsymbol{y}}_{k|k-1}\}. \quad (60)$$

Given that the Kalman innovation is uncorrelated with all the past observations and states (Katayama, 2006), $\tilde{\boldsymbol{y}}_{k|k-1}$ is uncorrelated with $\hat{\boldsymbol{x}}_{k|k-1}$, which means that we can simplify the above using *Lemma 5.4* from Katayama (2006) as

$$\hat{\boldsymbol{z}}_{k|k} = \mathbb{E}^*\big\{\boldsymbol{z}_k \mid \hat{\boldsymbol{x}}_{k|k-1}\big\} + \mathbb{E}^*\big\{\boldsymbol{z}_k \mid \tilde{\boldsymbol{y}}_{k|k-1}\big\}. \tag{61}$$

Substituting the first term from Theorem 1 and expanding $\boldsymbol{z}_k$ in the second term gives

$$\hat{\boldsymbol{z}}_{k|k} = \hat{\boldsymbol{z}}_{k|k-1} + \mathbb{E}^*\big\{(C_z\hat{\boldsymbol{x}}_{k|k-1} + \tilde{\boldsymbol{z}}_{k|k-1}) \mid \tilde{\boldsymbol{y}}_{k|k-1}\big\}. \tag{62}$$

Again, given that the innovation $\tilde{\boldsymbol{y}}_{k|k-1}$ is uncorrelated with the latent state estimate $\hat{\boldsymbol{x}}_{k|k-1}$, we get

$$\hat{\boldsymbol{z}}_{k|k} = \hat{\boldsymbol{z}}_{k|k-1} + \mathbb{E}^*\big\{\tilde{\boldsymbol{z}}_{k|k-1} \mid \tilde{\boldsymbol{y}}_{k|k-1}\big\} \tag{63a}$$

$$= \hat{\boldsymbol{z}}_{k|k-1} + \mathbb{E}\big\{\tilde{\boldsymbol{z}}_{k|k-1}\tilde{\boldsymbol{y}}_{k|k-1}{}^T\big\} \mathbb{E}\big\{\tilde{\boldsymbol{y}}_{k|k-1}\tilde{\boldsymbol{y}}_{k|k-1}{}^T\big\}^{-1} \tilde{\boldsymbol{y}}_{k|k-1}. \tag{63b}$$

$\square$

**Theorem 3.** *Given the state-space model of equation 1 and the associated Kalman filter equations 12-17, the linear minimum variance unbiased prediction of $\boldsymbol{z}_{k+l}$, for $l \geq 1$, given $\boldsymbol{y}_0 \dots \boldsymbol{y}_k$ is given by*

$$\hat{\boldsymbol{z}}_{k+l|k} \triangleq \mathbb{E}^*\{\boldsymbol{z}_{k+l} \mid \boldsymbol{y}_0 \dots \boldsymbol{y}_{k-1}, \boldsymbol{y}_k\} \tag{64a}$$

$$= C_z A^l \hat{\boldsymbol{x}}_{k|k-1} + \mathbb{E}\big\{\tilde{\boldsymbol{z}}_{k+l|k-1}\tilde{\boldsymbol{y}}_{k|k-1}{}^T\big\} \mathbb{E}\big\{\tilde{\boldsymbol{y}}_{k|k-1}\tilde{\boldsymbol{y}}_{k|k-1}{}^T\big\}^{-1} \tilde{\boldsymbol{y}}_{k|k-1}. \tag{64b}$$

*Proof.* By recursively replacing $\boldsymbol{x}_k^s$ from the first equation in equation 1, we can rewrite $\boldsymbol{z}_{k+l}$ as

$$\boldsymbol{z}_{k+l} = C_z\boldsymbol{x}_{k+l}^s + \boldsymbol{\epsilon}_{k+l} \tag{65a}$$

$$= C_z(A\boldsymbol{x}_{k+l-1}^s + \boldsymbol{w}_{k+l-1}) + \boldsymbol{\epsilon}_{k+l} \tag{65b}$$

$$= C_z(A(A\boldsymbol{x}_{k+l-2}^s + \boldsymbol{w}_{k+l-2}) + \boldsymbol{w}_{k+l-1}) + \boldsymbol{\epsilon}_{k+l} \tag{65c}$$

$$= \dots \tag{65d}$$

$$= C_z(A^l\boldsymbol{x}_k^s + A^{l-1}\boldsymbol{w}_k + \dots + A\boldsymbol{w}_{k+l-2} + \boldsymbol{w}_{k+l-1}) + \boldsymbol{\epsilon}_{k+l} \tag{65e}$$

$$= C_z(A^l\hat{\boldsymbol{x}}_{k|k-1} + A^l\tilde{\boldsymbol{x}}_{k|k-1} + A^{l-1}\boldsymbol{w}_k + \dots + A\boldsymbol{w}_{k+l-2} + \boldsymbol{w}_{k+l-1}) + \boldsymbol{\epsilon}_{k+l} \tag{65f}$$

$$= C_z A^l \hat{\boldsymbol{x}}_{k|k-1} + \underbrace{C_z(A^l\tilde{\boldsymbol{x}}_{k|k-1} + A^{l-1}\boldsymbol{w}_k + \dots + A\boldsymbol{w}_{k+l-2} + \boldsymbol{w}_{k+l-1}) + \boldsymbol{\epsilon}_{k+l}}_{=\tilde{\boldsymbol{z}}_{k+l|k-1}}. \tag{65g}$$

Now we can apply Lemma 2 to the above expression for $\boldsymbol{z}_{k+l}$ to show that

$$\hat{\boldsymbol{z}}_{k+l|k} = \mathbb{E}^*\{\boldsymbol{z}_{k+l} \mid \boldsymbol{y}_0 \dots \boldsymbol{y}_{k-1}, \boldsymbol{y}_k\} \tag{66a}$$

$$= \mathbb{E}^*\big\{\boldsymbol{z}_{k+l} \mid \hat{\boldsymbol{x}}_{k|k-1}, \tilde{\boldsymbol{y}}_{k|k-1}\big\}. \tag{66b}$$

Further simplifying, we get the linear minimum variance unbiased estimate of $\boldsymbol{z}_{k+l}$ given $\boldsymbol{y}_0 \dots \boldsymbol{y}_k$ as

$$\hat{\boldsymbol{z}}_{k+l|k} = \mathbb{E}^*\big\{\boldsymbol{z}_{k+l} \mid \hat{\boldsymbol{x}}_{k|k-1}, \tilde{\boldsymbol{y}}_{k|k-1}\big\} \tag{67a}$$

$$= \mathbb{E}^*\big\{\boldsymbol{z}_{k+l} \mid \hat{\boldsymbol{x}}_{k|k-1}\big\} + \mathbb{E}^*\big\{\boldsymbol{z}_{k+l} \mid \tilde{\boldsymbol{y}}_{k|k-1}\big\} \tag{67b}$$

$$= \mathbb{E}^*\big\{C_z A^l\hat{\boldsymbol{x}}_{k|k-1} + \tilde{\boldsymbol{z}}_{k+l|k-1} \mid \hat{\boldsymbol{x}}_{k|k-1}\big\} \tag{67c}$$

$$+ \mathbb{E}^*\big\{C_z A^l\hat{\boldsymbol{x}}_{k|k-1} + \tilde{\boldsymbol{z}}_{k+l|k-1} \mid \tilde{\boldsymbol{y}}_{k|k-1}\big\} \tag{67d}$$

where we have used the same simplification techniques as in Theorems 1-2. Given that $\tilde{\boldsymbol{z}}_{k+l|k-1}$ is uncorrelated with $\hat{\boldsymbol{x}}_{k|k-1}$, and $\hat{\boldsymbol{x}}_{k|k-1}$ is uncorrelated with $\tilde{\boldsymbol{y}}_{k|k-1}$, we have

$$\hat{\boldsymbol{z}}_{k+l|k} = C_z A^l \hat{\boldsymbol{x}}_{k|k-1} + \mathbb{E}^*\big\{\tilde{\boldsymbol{z}}_{k+l|k-1} \mid \tilde{\boldsymbol{y}}_{k|k-1}\big\} \tag{68a}$$

$$= C_z A^l \hat{\boldsymbol{x}}_{k|k-1} + \mathbb{E}\big\{\tilde{\boldsymbol{z}}_{k+l|k-1}\tilde{\boldsymbol{y}}_{k|k-1}{}^T\big\} \mathbb{E}\big\{\tilde{\boldsymbol{y}}_{k|k-1}\tilde{\boldsymbol{y}}_{k|k-1}{}^T\big\}^{-1} \tilde{\boldsymbol{y}}_{k|k-1}. \tag{68b}$$

$\square$

Theorems 2-3 derive the optimal linear minimum variance unbiased filtered and forward predicted estimates of $\boldsymbol{z}$ in terms of the second order statistics of the data residuals. We will next show that these second order statistics can be obtained in an asymptotically unbiased manner using the solution to equation 51.

**Theorem 4.** *Given the state-space model of equation 1 and the associated Kalman filter equations 12-17, the linear minimum variance unbiased prediction of $\boldsymbol{z}_k$ given $\boldsymbol{y}_0 \ldots \boldsymbol{y}_k$ is given by equation 6, where $C_z K_f$ is taken as the asymptotic solution to equation 51 (i.e., the general form of equation 7).*

*Proof.* Looking at the right hand side of equation 51, we first note that each term is correlated with a sample covariance. First, columns of $\tilde{Y}$ consist of samples of the innovation sequence $\tilde{\boldsymbol{y}}_{k|k-1}$, thus $\frac{1}{N}\tilde{Y}\tilde{Y}^T$ asymptotically converges to the covariance of the innovation sequence:

$$\lim_{N\to\infty} \frac{1}{N}\tilde{Y}\tilde{Y}^T = \mathbb{E}\left\{\tilde{\boldsymbol{y}}_{k|k-1}\tilde{\boldsymbol{y}}_{k|k-1}^T\right\}. \tag{69}$$

Similarly, block row number $l+1$ (i.e., rows $ln_z + 1$ to $ln_z + n_z$) in $\tilde{Z}_{1:i}$, which is denoted by $\tilde{Z}_{l:l+1}$, consists of samples of the residual secondary signal predictions $\tilde{\boldsymbol{z}}_{k+l|k-1}$. Thus the associated block row of $\frac{1}{N}\tilde{Z}_{l:l+1}\tilde{Y}^T$ asymptotically converges to the following cross covariance:

$$\lim_{N\to\infty} \frac{1}{N}\tilde{Z}_{l:l+1}\tilde{Y}^T = \mathbb{E}\left\{\tilde{\boldsymbol{z}}_{k+l|k-1}\tilde{\boldsymbol{y}}_{k|k-1}^T\right\}. \tag{70}$$

Specifically, the first $n_z$ rows of this matrix converge to

$$\lim_{N\to\infty} \frac{1}{N}\tilde{Z}_{0:1}\tilde{Y}^T = \lim_{N\to\infty} \frac{1}{N}\tilde{Z}\tilde{Y}^T = \mathbb{E}\left\{\tilde{\boldsymbol{z}}_{k|k-1}\tilde{\boldsymbol{y}}_{k|k-1}^T\right\}. \tag{71}$$

Replacing equations 69-71 in equation 51 and comparing against the optimal estimation equation 58 (and 64a), establishes that the $C_z K_f$ learned asymptotically via equation 51 yields the minimum variance unbiased prediction of $\boldsymbol{z}_k$. $\qquad\square$

### A.8.4 SOLVING FOR THE EXACT SOLUTION FOR $K_f$

Previously, we showed how $C_z K_f$ can be identified, and that solution is always available. We also explained why as far as the practical problem of predicting/filtering behavior is concerned, identifying $C_z K_f$ is sufficient and we do not need to identify $K_f$ separately. Here, we will discuss the conditions under which $K_f$ itself is also identifiable, which is not always the case. This is fundamentally because not all latent states are always relevant to behavior. More formally, the pair $(C_z, A)$ is not always observable, which means that even when we observe the secondary signal $\boldsymbol{z}_k$, the latent states $\boldsymbol{x}_k$ are not always fully observable. Thus, for these systems, even in the PSID setting where we observe $\boldsymbol{z}_k$, the $K_f$ associated with certain latent states is not uniquely identifiable. For example, consider the special case of a system where $\boldsymbol{z}_k = \boldsymbol{y}_k$. In such a system, the PSID result would be the same as the regular subspace identification result, and the $K_f$ would thus not be uniquely identifiable.

However, in the special case where all latent states are relevant to the secondary signal (i.e., the pair $(C_z, A)$ is observable), $C_z K_f$ can be decomposed into updated $C_z$ and $K_f$ matrices. An example of this would be any time when $C_z$ has a left pseudo-inverse. In this case, multiplying the computed $C_z K_f$ by that left pseudo-inverse would give us the exact solution for $K_f$ that optimizes the filtering of the secondary signal. Similarly, in the more general formulation from the previous section, whenever $\Gamma_z$ has a left pseudo-inverse, multiplying the computed $\Gamma_z K_f$ by that left pseudo-inverse would give us the exact solution for $K_f$.

Since in general this solution is not available, in this new method, which we call *PSID with filtering*, we always only learn $C_z K_f$ from the data and use that in generating our filtered estimate of the secondary signal.

### A.8.5 SKETCH OF THE PROOF FOR PSID WITH SMOOTHING [ADDED IN REVISION]

In our results (Figure 2c), we empirically validated that PSID with smoothing reaches the performance of ideal smoothing (i.e., with true model parameters). Here, we provide a sketch of the proof for why

PSID with smoothing gives optimal (in the sense of minimum variance unbiased) smoothed estimates of $z_k$ given $\boldsymbol{y}_0 \dots \boldsymbol{y}_N$.

To review, in PSID with smoothing, we apply PSID with filtering twice. First, we apply PSID with filtering normally, in the forward direction. Second, we compute the residuals of the secondary signal ($\tilde{\boldsymbol{z}}_{k|k}$) and apply PSID with filtering to those residuals in the backward time direction.

To prove that the aforementioned procedure results in optimal smoothed estimates, we assume that PSID with filtering asymptotically (i.e., with enough training data) achieves optimal filtered estimates of the secondary signal. Thus, the predictions obtained from the forward pass, i.e., $\hat{\boldsymbol{z}}_{k|k}$, give the expected value of $z_k$ given $\boldsymbol{y}_0 \dots \boldsymbol{y}_k$. Moreover, the residual $\tilde{\boldsymbol{z}}_{k|k}$ will be orthogonal to the subspace spanned by $[\boldsymbol{y}_0 \dots \boldsymbol{y}_k]$.

In the backward pass, PSID with filtering is applied to $[\boldsymbol{y}_N \dots \boldsymbol{y}_0]$ and $\tilde{\boldsymbol{z}}_{N|N} \dots \tilde{\boldsymbol{z}}_{0|0}$. We denote the final predictions of this backward model as $\hat{\tilde{\boldsymbol{z}}}_{k|k}$ (equation 9). Again, due to the assumed optimality of PSID with filtering, $\hat{\tilde{\boldsymbol{z}}}_{k|k}$ gives the expected value of $\tilde{\boldsymbol{z}}_{k|k}$ given $\boldsymbol{y}_N \dots \boldsymbol{y}_k$. Let's denote the error of this backward prediction as $\tilde{\tilde{\boldsymbol{z}}}_{k|k}$. $\tilde{\tilde{\boldsymbol{z}}}_{k|k}$ will be orthogonal to the subspace spanned by $[\boldsymbol{y}_N \dots \boldsymbol{y}_k]$.

Overall, the forward pass effectively projects $\boldsymbol{z}_k$ onto $\boldsymbol{y}_0 \dots \boldsymbol{y}_k$ and the backward pass projects the residual of the first projection onto $\boldsymbol{y}_N \dots \boldsymbol{y}_k$. As such, any final residual $\tilde{\tilde{\boldsymbol{z}}}_{k|k}$ should fall outside the subspace of the entire observed timeseries, i.e., $\boldsymbol{y}_0 \dots \boldsymbol{y}_N$, which means that it will be uncorrelated with the data. Assuming Gaussian distributions for $\boldsymbol{\epsilon}_k$ and thus $\boldsymbol{z}_k$, this means that the residual will be independent of $\boldsymbol{y}_0 \dots \boldsymbol{y}_N$ and thus cannot be estimated by the data. In other words, the final estimate $\hat{\boldsymbol{z}}_{k|N}$ from equation 9 is the optimal prediction given $\boldsymbol{y}_0 \dots \boldsymbol{y}_N$.

### A.8.6 RELATIONSHIP OF PSID WITH SMOOTHING AND THE KALMAN SMOOTHER
[ADDED IN REVISION]

One may wonder if there is any evidence that the general forward-backward filtering structure used in PSID with smoothing could yield optimal smoothing estimates given ideal model parameters. To answer that, we refer the reader to our inspiration for this architecture: the forward-backward two-filter formulation for the Kalman smoother (equations 33-36), which consists of two Kalman filters running in the forward and backward directions (Fraser & Potter, 1969; Kitagawa, 2023), in a manner akin to PSID with smoothing. While the learned forward and backward models in PSID with smoothing can be used to run non-steady-state Kalman filters in forward and backward directions, to establish the relationship with the Kalman smoother, we will start from the steady-state formulation.

We showed in Appendix A.5.1 that the steady state optimal Kalman smoothed state is a fixed weighted average between the forward and backward filters, per equation 44. At a high level, given ideal model parameters, PSID with smoothing would implement a similar two-filter Kalman smoother. We only make one simplifying, but potentially non-essential, deviation from the Kalman smoother formulation in our implementation that we explain below.

Multiplying both sides of equation 44 by $C_z$, we can conclude that the optimal Kalman smoothed estimate of the secondary signal $\boldsymbol{z}_k$ at steady-state is

$$\hat{\boldsymbol{z}}_{k|N} = \underbrace{C_z P_s P^{-1} \hat{\boldsymbol{x}}_{k|k}}_{\text{Forward PSID pass}} + \underbrace{C_z P_s \hat{\boldsymbol{x}}^b_{k|k+1}}_{\text{Backward PSID pass}} \ . \tag{72}$$

Intuitively, the forward pass in PSID aims to learn the first term, and the backward pass aims to learn any residual, i.e., the second term. As shown in equation 9, for simplicity, we do not learn the weight for adding the forward estimations to the backward estimations. This amounts to assuming $P_s P^{-1} = \left(P^{-1} + P^b_p\right)^{-1} P^{-1} \approx I$, which still gives empirically almost ideal smoothing accuracy in our simulations (Figure 2c). Nevertheless, if desired, the ideal weight to scale the forward estimates before combining with the backward estimates can be obtained empirically via a regression in the training data.

## A.9 DPAD PREDICTOR FORM FORMULATION [ADDED IN REVISION]

In the special case of all DPAD parameters being linear matrix multiplications, the DPAD model (equation 10) reduces to the predictor form representation of a linear state space model (equation 4). As explained in Section 3.2, the predictor form is indeed based on the Kalman filter; that is, the predictor form is the representation of the system that takes the Kalman predicted latent states as the latent states. In other words, in the special case of linear modeling, DPAD can numerically learn and implement an optimal Kalman predictor. Sani et al. (2024) indeed show in their *Extended Data Fig. 1* that in this linear case, DPAD reaches the same performance as an ideal Kalman filter with true model parameters.

The linear setting is a great example of how DPAD could implicitly handle noise covariances, etc. in the data to achieve optimal prediction. Note that the steady state Kalman filter recursions can be written as equation 3, which we repeat below for easier exposition:

$$\hat{\boldsymbol{x}}_{k+1|k} = (A - KC_y)\hat{\boldsymbol{x}}_{k|k-1} + K\boldsymbol{y}_k. \tag{73}$$

This simple equation considers uncertainties in observations and state estimates optimally and aggregates all of that information into the fixed matrix $K$, which is the steady state Kalman gain. DPAD can implement an optimal Kalman filter at steady-state by learning the final steady state Kalman gain $K$, along with the other parameters of the above equation, numerically via gradient descent. In other words, DPAD can implicitly track uncertainty at steady-state as is, by directly learning (via numerical optimization) the inference recursions that yield the best prediction (lower MSE) in the training data. A similar argument can be made to see how DPAD with smoothing could implement the steady-state behavior of a Kalman smoother (Appendix A.5.1) via its bidirectional RNN.

Beyond steady-state, given additional latent states, the RNN architecture of DPAD is also capable of modeling some time-varying system dynamics, by learning to track the time-varying system parameters via the additional latent states. This is because RNNs are universal approximators that can implement any state-space dynamics, under mild conditions (Schäfer & Zimmermann, 2006). For example, if the $A$ parameter itself evolves as a random walk or some other recursive evolution in the form of equation 1, then a *nonlinear* RNN with additional states can learn the dynamics of $A$ via a subset of states and apply the current estimated $A$ onto the other latent states via its overall nonlinear state transition function. This capacity could, for example, allow the RNN to *approximate* the time-varying Kalman gain in a Kalman filter *before* it reaches steady state. Specifically, as long as the optimal time-varying Kalman gain can be written as, or approximated as, a recursive state-space model by an RNN, DPAD should be able to learn to implement that time-varying Kalman filter by including additional latent states that track the time-varying Kalman gain. However, the model learned by the RNN may still only approximate a time-varying Kalman filter, because it is not clear if neural networks can learn an accurate solution for the matrix inversion required in the optimal time-varying Kalman gain equation 13c; rather, the model may learn something akin to a Taylor series expansion of the matrix inversion. Finally, it is important to note that given the numerical nature of the learning in DPAD, additional dynamics such as time-varying model parameters would only be learned if they meaningfully reduce the optimization loss (i.e., improve the prediction accuracy) in a meaningful part of the training data. The time-varying Kalman filter, for example, would likely only have a meaningful advantage over the steady-state version if the data consists of many short sequences, where the state estimation repeatedly restarts from the initial state. The setup of the Neural Latents Benchmark is one such example; in contrast, modeling a complete recording session as a single continuous segment (as we do for the Sabes data) is unlikely to benefit from time-varying Kalman filtering.

## A.10 SABES DATASET DETAILS

The Sabes dataset consisted of recordings from a non-human primate controlling a cursor based on fingertip position on a 2D surface in a 3D virtual reality environment. The behavior signal was the 2D cursor position and velocity. Neural activity from M1 was recorded using a 96-electrode microelectrode array. We selected a random subset of 30 of these electrodes for our analysis.

We extracted three neural modalities: spikes, raw local field potential (LFP), and local field potential power bands. For spikes, we counted spikes in non-overlapping 10-ms bins, smoothed with a Gaussian

kernel with standard deviation 50 ms, and then downsampled to a 50-ms time-step. The LFP features were extracted from these same 30 electrodes. We modeled both raw LFP and LFP log-powers using the processing as described in Sani et al. (2024). We analyzed data from seven sessions (20160622/01 through 20160921/01).

All neural features and behavior are z-scored using training statistics. For cross-validation, we use session-wise folds; unless otherwise noted, decoding metrics are computed on held-out test segments with no temporal overlap. We report causal (prediction, filtering) and smoothed decoding using identical splits and hyperparameters across conditions.

### A.11 OTHER BASELINE METHODS FROM THE NLB

The following is a brief summary about the methods from the NLB leader-board that we include in Table 1 but were not discussed in the main text:

**MINT** (Perkins et al., 2024): an interpretable decoder that represents neural activity via a library of canonical neural trajectories and interpolates within this mesh to decode behavior.

**LangevinFlow** (Song et al., 2025): a sequential VAE whose latent dynamics follow underdamped Langevin equations, combining RNN encoders and Transformer decoders to capture oscillatory, flow-like neural population dynamics.

**GPFA** (Yu et al., 2009): extracts smooth, low-dimensional latent trajectories from neural population activity by combining the smoothing and dimensionality reduction operations in a common probabilistic framework.

**SLDS** (Linderman et al., 2017): models neural data using switching linear dynamical systems, where latent dynamics transition between multiple linear regimes, enabling segmentation of nonlinear activity into interpretable dynamical modes.

**NDT** (Ye & Pandarinath, 2021): adapts the Transformer architecture to neural population data, replacing recurrent architectures and modeling temporal dependencies in spiking activity without explicit latent state assumptions.

**NDT3** (Ye et al., 2024): a foundation model for motor decoding, and a scaled-up version of NDT, pre-trained on data from multiple labs, enabling strong generalization across tasks and neural distribution shifts.

### A.12 MODEL TRAINING AND HYPERPARAMETER SELECTION FOR DPAD VARIANTS

For our submissions to the NLB, we performed a comprehensive hyperparameter search for DPAD with smoothing and DPAD with smoothing with ensemble. All choices were made based on performance on the provided validation splits for each dataset and task.

**Hyperparameter search space**  We searched over key parameters that control model capacity, architecture, and optimization. The primary recurrent neural network used for the state recursion ($A$) was a long short-term memory (LSTM). For the behavior decoding readout ($C_z$), we tested MLPs with one or two hidden layers. For the held-out neural prediction task (co-smoothing), the readout network ($C_z$) was an MLP with an exponential output activation to model firing rates. Table 2 summarizes the search space.

Table 2: Hyperparameter search space used on the validation split of NLB datasets for DPAD variants.

| Component | Hyperparameter | Values Searched |
|---|---|---|
| Optimization | Learning Rate | {1e-4, 1e-3} |
| | Weight Decay (AdamW) | {1, 1e-1, 1e-2, 1e-3} |
| | Optimizer | AdamW |
| Architecture | Latent Dimension ($n_x$) | {64, 256, 512, 1024} |
| | State Recursion ($A$) | LSTM |
| | Behavior Readout ($C_z$) | 1 or 2 hidden layers; {256, 512} units |

**Ensemble selection**   For the ensembled DPAD with smoothing models, we followed a two-stage validation procedure. First, we trained a pool of models by sampling from the hyperparameter space outlined in Table 2. Second, we used the validation set to determine the optimal number of models to include in the ensemble. We iteratively created ensembles of increasing size by averaging the predictions of the top-performing models, and selected the ensemble size that maximized validation performance. This process was performed independently for the behavior decoding and held-out neural prediction tasks for each dataset.

## A.13   COMPUTATIONAL COST AND REAL-TIME CONSIDERATIONS [ADDED IN REVISION]

In this subsection, we provide a discussion of the real-time applicability and the computational cost associated with both training and inference for the proposed filtering and smoothing methods.

**Inference cost**   For both PSID and DPAD, the inference cost of the smoothing variants introduced in this work is higher than that of their causal (forward-only) counterparts, since smoothing performs a full latent-state inference pass twice: once in the forward direction and once in the backward direction. As a numerical example, on a desktop computer with an Intel Core i7-9700K @ 3.60 GHz CPU, with a latent state dimension of $n_x = 32$ on the NLB data, causal PSID takes $0.0061$ ms per time-step, PSID with Smoothing takes $0.0124$ ms per time-step, while causal DPAD takes $0.210$ ms per time-step, and DPAD with smoothing takes $0.254$ ms per time-step.

**Training cost**   For PSID, the training cost of the smoothing variant is slightly more than doubled relative to causal PSID, since smoothing applies PSID in both the forward and backward directions and includes an additional reduced-rank regression step. For DPAD, the training cost is not deterministic, as the model is learned via numerical optimization. Empirically, however, under identical settings (dataset size, latent dimension, number of epochs, etc.), we observed that the average training time of DPAD with smoothing is 1.32 times that of DPAD. For instance, on the MC_Maze dataset with latent dimension $n_x = 32$, training DPAD with smoothing for 100 epochs required 107.33 seconds. When searching over different sets of hyperparameters and nonlinearities, all options can be trained in parallel, so the order of magnitude of the time required does not change.

**Real-time applicability**   PSID and DPAD both perform inference through recurrent latent-state updates, making their forward (causal) inference computationally efficient and suitable for real-time deployment. The smoothing extensions inherit this computational efficiency. However, these smoothing versions are inherently offline methods. This is not due to computational cost, but because, by definition, smoothing requires access to future data.

Nevertheless, in real-time applications where some fixed delay of $d$ samples is acceptable, one could still use these computationally efficient smoothing methods by processing the data in a sliding window and always returning the smoothed estimates from $d$ samples before the latest sample in the window – this is because even smoothing is highly computationally efficient, for example, PSID takes $0.0124$ ms per time-step and DPAD takes $0.254$ ms per time-step, which is much shorter than typical bins used to process neural data in brain-computer interfaces ($\sim$ 10-50 ms bins).

## A.14   VALIDATION OF PSID WITH FILTERING

As discussed in Section 4.1, Figure 5 shows the error of PSID with filtering in learning the model parameters as the number of training samples increases. To compare learned parameters with ground truth parameters for a given model, we first use the method presented in (Sani et al., 2021) to change the basis of the learned model via a similarity transform to one that is aligned with that of the true model. This does not change the learned model, but makes the learned parameters comparable to the true parameters. We then compute the Frobenius norm of the difference between the learned and true parameters, normalized by the Frobenius norm of the true parameters. We compute this metric for all main model parameters learned by the original PSID method (i.e., $A$, $C_y$, $C_z$, $K$, $\Sigma_y$), as well as the additional $C_z K_f$ parameter learned in this work for PSID with filtering.

$K_f$ is an example of a non-identifiable parameter, for which as expected the error does not converge to zero (Figure 5). For the random models in this simulation, state and observation dimensions were chosen randomly, so for many systems $n_z < n_x$ and the pair $(C_z, A)$ was not observable, which

means that $K_f$ was not uniquely identifiable (see Appendix A.8.4). Note that even though $K_f$ is an internal characteristic and not in general learnable (Appendix A.8.4), $C_z K_f$ which is relevant for filtering is accurately learned (Figure 5).

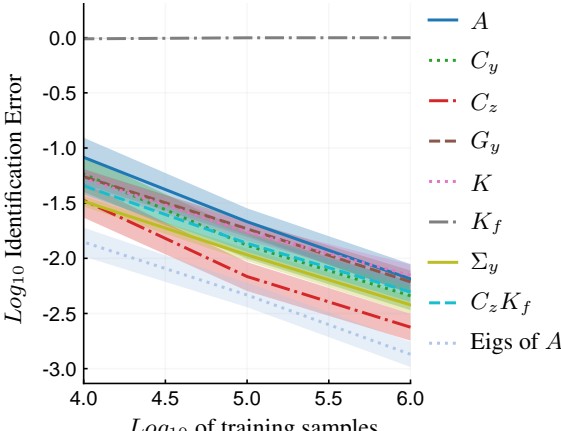

Figure 5: The learned parameters, including the $C_z K_f$ learned for filtering, converge to the ground truth values with increasing training samples. The error for each parameter is computed as the Frobenius norm of the difference between the learned and true value, normalized by the Frobenius norm of the true value of that parameter matrix. Solid lines show the mean error across the 20 simulated models, and shaded areas show the standard error of the mean (s.e.m.). For all identifiable parameters, the mean error converges to below 1% with 1 million training samples.

### A.15 PSID CANNOT BE EXTENDED TO FILTERING BY JUST SHIFTING THE TRAINING BEHAVIOR DATA

Ultimately, PSID optimizes the one-step-ahead prediction of the secondary signal using the primary signal. One might ask: if we shift the behavior signal one step forward in time during training, wouldn't that simply result in optimal filtering of the secondary signal? This is an interesting idea, and indeed it improves the filtering performance of the secondary signal using PSID. However, as we show in this section, the optimal filter in the general case where state and observation noises are correlated (that is, $S \neq 0$ in equation 2) is not a simple shifted predictor and requires a two-step filtering and update procedure during filtering.

We also empirically compare the decoding performance (R2) for the shifted PSID as a baseline and show that while this baseline does improve the estimation accuracy over ideal one-step-ahead prediction, it does not reach the filtering performance of the true models, whereas PSID with filtering does achieve optimal performance (Figure 6).

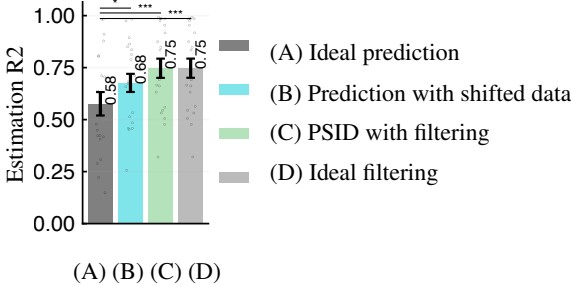

Figure 6: Comparison of the performance of the shifted PSID and PSID with filtering. While PSID with shifted data outperforms even an ideal (ground truth) 1-step ahead prediction, it does not reach ideal filtering accuracy, whereas the new PSID with filtering method reaches ideal filtering accuracy.

We can see why this shifted-data PSID approach cannot reach optimal filtering by inspecting the Kalman filter equations (Appendix A.2). The original PSID method identifies the parameters of the predictor form of a state-space model, including the predictor gain $K$ (equation 13c). This is sufficient for one-step-ahead prediction (equation 12b). However, optimal filtering requires the update step in equation 12a, which uses the filter gain $K_f$. As shown in equation 13c, the total gain is $K = AK_f + K_v$. When $S \neq 0$, $K_v$ (equation 13b) is non-zero, and thus $K_f$ cannot be uniquely determined from the predictor parameters $A$ and $K$. Since the standard PSID procedure does not identify $K_f$, it cannot produce an optimal filtered estimate in the general case.

### A.16    ALTERNATIVE BACKWARD MODEL FOR PSID SMOOTHING

As noted in the main text, the backward model learned for smoothing in our proposed method is different from the backward representation of the underlying stochastic model as formulated in Figure 3.5 of Van Overschee & De Moor (1996) and Appendix A.4. We empirically demonstrate this distinction here. The primary difference lies in the data used to train the backward model. In our proposed PSID smoothing approach, the backward model is learned using the *residual* of the secondary signal $\tilde{z}_k$, i.e., the portion of the secondary signal not explained by the forward PSID model. An alternative approach would be to train the backward model on the time-reversed secondary signal $z_k$ itself (see Section 3.4.3). As we validate here, this alternative procedure indeed learns the backward stochastic model from Van Overschee & De Moor (1996) (see Appendix A.4).

Figure 7 presents an empirical comparison of these two alternative backward passes. We simulated data from models with random parameters, and compared the learned parameters for the backward PSID model with the parameters of the backward stochastic form representation of the true model. Figure 7a shows the difference between the parameters learned with the alternative method (using secondary signal itself in the backward pass) with the parameters of the backward stochastic form. The identified parameters indeed converge increasingly closer to the parameters of the backward stochastic form. In contrast, as shown in Figure 7b, our proposed method, which uses the residuals of the forward filter, learns a different backward model (as expected) that is further away from the backward stochastic form. This model is tailored to explaining the errors of the forward pass, leading to superior (as high as ideal) smoothing performance as shown in the main text (Figure 2c).

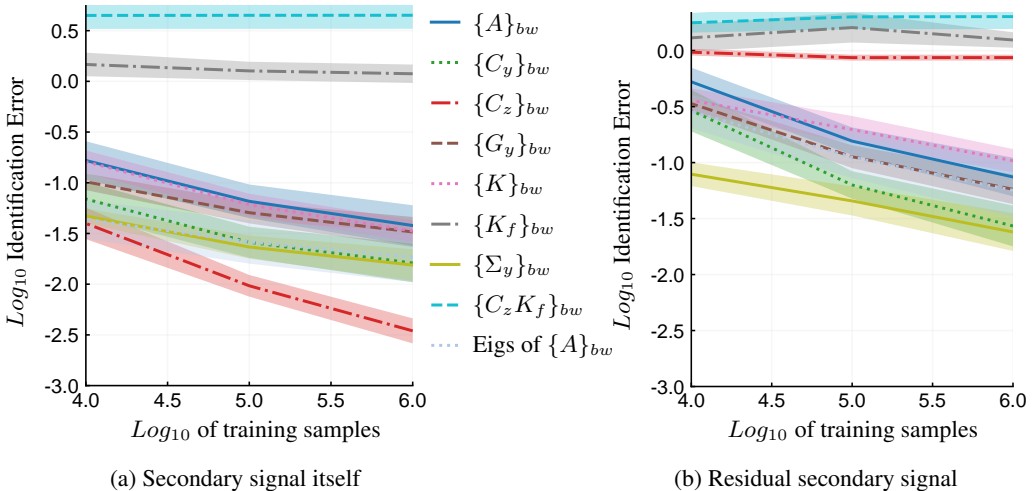

(a) Secondary signal itself          (b) Residual secondary signal

Figure 7: The relation between the learned backward PSID model and the backward stochastic form of the state space model. The normalized difference between the learned parameters of the backwards model and the backward stochastic form when (**a**) the secondary signal itself is used to learn the backward model, or (**b**) the residual secondary signal is used to learn the backward model. The latter case as expected is further away from the backward stochastic form.

## A.17 LINEAR BEHAVIOR DECODING ACROSS STATE DIMENSIONS

As reported in Section 4.3 we found that smoothing achieved higher accuracy than filtering, which achieved higher accuracy than one-step-ahead prediction for both PSID and NDM across state dimensions (Figure 8). Additionally, across state dimensions, PSID outperformed its NDM counterparts.

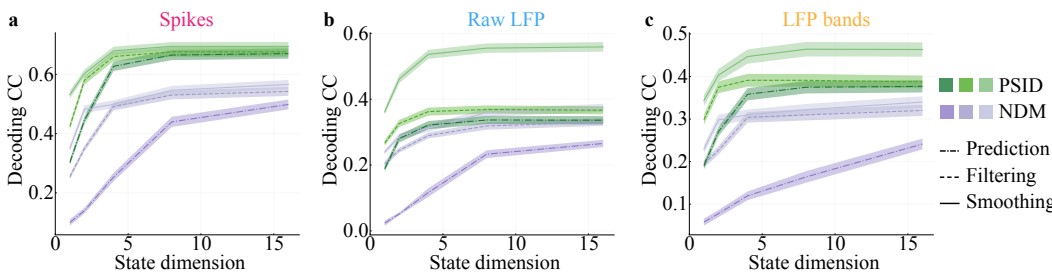

Figure 8: **(a)** Cross-validated decoding accuracy correlation coefficient (CC) achieved by PSID and NDM for prediction, filtering, and smoothing across different latent state dimensions. Lines show the average across sessions and folds ($N = 35$ session-folds), and the shaded areas show the s.e.m. The neural modality is population spiking activity. **(b)** Same as **(a)** but for raw LFP activity. **(c)** Same as **(a)** but for LFP power band activity.

## A.18 EFFECT OF RECURRENT CELL TYPE IN CAUSAL DPAD AND DPAD WITH SMOOTHING
[ADDED IN REVISION]

The recurrent cell used in DPAD is a flexible design choice with both vanilla RNN cells and LSTM cells already supported in the code (Sani et al., 2024). We used LSTM cells in our NLB results in Table 1, but here provide results for other cells. Figure 9 provides a direct comparison of vanilla RNN and LSTM cell types on the NLB validation splits. Across most latent state dimensions and datasets, the LSTM cell achieves higher performance for both behavior decoding and held-out neural prediction.

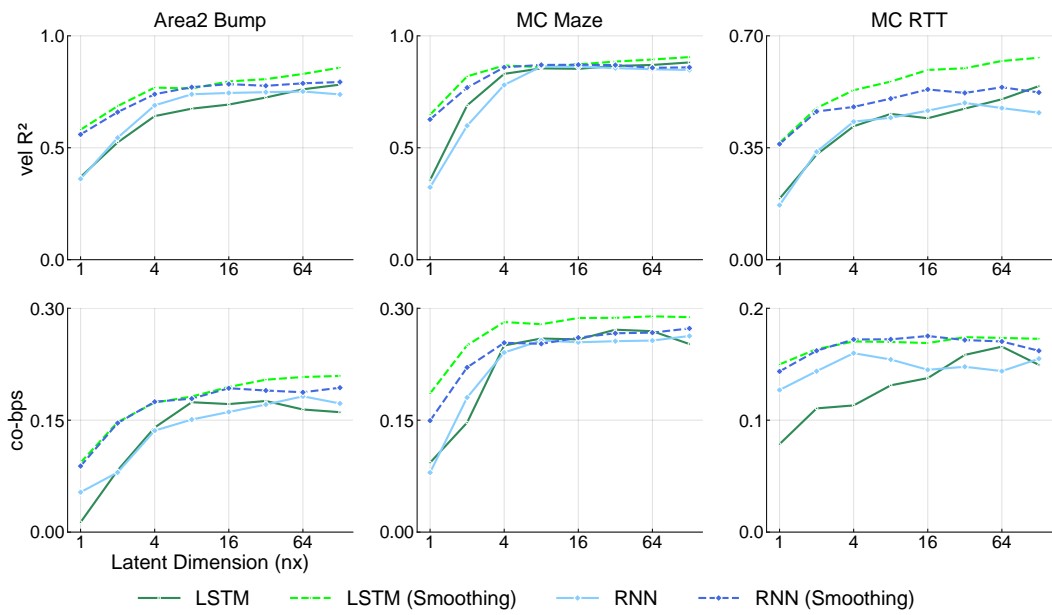

[added in revision] Figure 9: Comparison of LSTM vs. RNN cells for DPAD with smoothing and causal DPAD on the NLB datasets. Performance is evaluated on the validation split for both behavior decoding ($R^2$, top panels) and held-out neural prediction (co-bps, bottom panels) across latent state dimensions. The LSTM cell (used in our NLB submissions reported in Table 1) consistently achieves better smoothing results than the RNN across most state dimensions and datasets.

