# OpenReview forum: "Preferential dynamic modeling with forward-backward smoothing"
_ICLR.cc/2026/Conference — Submitted to ICLR 2026_

### Official Review · Reviewer_55zD · 2025-10-31

**Soundness:** 2
**Presentation:** 3
**Contribution:** 3
**Rating:** 6
**Confidence:** 2

**Summary:**

This paper extends PSID (linear) and DPAD (nonlinear) to support both filtering and smoothing of a secondary signal from a primary time series, thereby bridging causal and non-causal inference in two-signal system identification. The authors provide analytical derivations for the linear case and introduce a bidirectional RNN variant for the nonlinear case. Experiments on simulations, primate motor cortex data, and the Neural Latents Benchmark (NLB) demonstrate performance competitive with top-ranking methods such as Ctrl-TNDM and LangevinFlow, achieving top behavior-decoding accuracy on the MC_RTT dataset.

**Strengths:**

- Well written, clearly structured, and easy to follow.

- Strong empirical validation across multiple datasets and settings, with consistent improvements over relevant baselines.

**Weaknesses:**

- Theoretical contributions are presented in a textbook style, making it hard to separate novel insights from background material. Appendix e.g. lacks formal structuring (e.g., propositions or theorems) that would clarify assumptions and originality.

**Questions:**

- Can the authors summarize their key theoretical contributions more formally (e.g., in a compact proposition) to distinguish them from standard Kalman filtering results?

---

> ### Author Response · Authors · 2025-11-22
>
> We thank the reviewer for their time and valuable feedback. We have now added multiple appendices to formally present proofs for our derivations (**Appendices A.8.3, A.8.5, A.8.6**). These sections include formal theorems that clarify our assumptions and distinguish our results from background Kalman information. The reason we include the background sections on Kalman and system identification is both to help with our derivations and to provide a helpful reference for the community.
>
> Finally, the following is a summary of our novel insights and original contributions. Overall, our extensions of PSID provide the optimal filtering and smoothing parameters in the two-signal setting, unlike the original PSID, which only provides optimal parameters for one-step-ahead prediction. Similarly, our extension of DPAD enables smoothing, whereas the original only supported one-step ahead prediction. To enable the above, we made the following novel contributions:
>
> ---
>
> 1. **Novelties on the linear modeling side, the filtering problem (inferring $z_k$ given $y_1$ to $y_k$)**:
>
>     **a)** The theoretical insight that in the two-signal setting, more of the latent dynamics become “observable” and thus identifiable, and how this enables useful filtering/smoothing in contrast to standard one-signal latent state space modeling.
>
>     **b)** The theoretical insight that because of the above, parameters like $K_f$ (the Kalman gain for the Kalman update equation 12a) that are not uniquely identifiable in the standard one-signal setting, become partially identifiable in the two-signal setting.
>
>     **c)** The derivation that filtering (k|k) estimation of the secondary signal can be done if $C_z K_f$ is identified, and does not require full identification of $K_f$.
>
>     **d)** A method for identifying $C_z K_f$ via reduced-rank regression (RRR) and thus enabling filtering (k|k) estimation of the secondary signal. As we now formally prove in the **new Appendix A.8.3**, this method achieves optimal (minimum-variance unbiased) filtering.
>
>     **e)** The numerical validation that our method reaches ideal filter (k|k) estimation performance (Figure 2b).
>
>     **f)** The demonstration that simply shifting the secondary signal by one sample is not sufficient for allowing PSID to achieve the prediction accuracy that we achieve with the above method (Appendix A.15 and Figure 6).
> ---
>
> 2. **Novelties on the linear modeling side, the smoothing problem (inferring $z_k$ given $y_1$ to $y_N$)**:
>
>     **a)** The theoretical insight that, similar to the forward-backward (two-filter) smoother formulation for Kalman smoothing, one could directly learn the forward and backward filters from data (in the Kalman smoother, the backward filter is parameterized based on the model parameters of the forward filter).
>
>     **b)** The insight that learning a backward model on the total secondary signal correctly learns the parameters of the backward stochastic model (Figure 7a), but it will not be ideal in terms of predicting the secondary signal.
>
>     **c)** The insight that, instead of b, learning a backward model for the residual secondary signal — which, as expected, learns a model that is different from the backwards stochastic model (Figure 7b) — reaches ideal smoothing performance (Figure 2c). We now provide a sketch of the proof for this in the **new Appendix A.8.5**.
> ---
>
> 3. **On the nonlinear modeling side**:
>
>     **a)** The DPAD paper shows how in the special case of linear models, DPAD achieves the same objectives/performance as PSID, but with numerical optimization. Here, our key novelty is to identify that changing the RNN cell in DPAD to be bidirectional would conceptually be equivalent to our two-filter forward-backward generalization of PSID for smoothing. We implement this and show that it indeed yields very competitive results in various datasets, including outperforming the SOTA results in the MC_RTT dataset on the NLB leaderboard.

---

### Official Review · Reviewer_24bz · 2025-11-01

**Soundness:** 2
**Presentation:** 2
**Contribution:** 2
**Rating:** 2
**Confidence:** 3

**Summary:**

In this work, the paper proposes the framework extensions of two existing dual-signal latent state-space modeling methods. The first is a linear method named Preferential Subspace Identification (PSID), and the second one is the non-linear Dynamic Prioritized Analysis of Dynamics (DPAD). The goal of this proposed framework is to enable optimal filtering and smoothing of a secondary signal e.g., behavior, $z_k from the primary neural activities. As for PSID, the authors use the Reduced Rank Regression (RRR) step to do the filtering extension. They also extend the PSID with forward-backward smoothing. Then for the DPAD, they use the bidirectional RNN (Bi-RNN) deep model architecture to implement the smoothing. By comparison, the original PSID and DPAD methods are focused on one-step-ahead prediction.

For the experiment parts, the authors validate their proposed framework with simulation data and the widely-adopted NLB benchmark.

**Strengths:**

1. The paper writing flow is concrete and easy-to-follow.
2. The paper provides a comprehensive framework addressing the filtering and smoothing tasks in two wide-adopted models, linear (PSID) and nonlinear (DPAD). These problems are actually quite important in computational neural data modeling.

**Weaknesses:**

1. The overall framework in the submitted manuscripts is like a combination of the incremental model enhancement of two existing methods, which is pretty heuristic and intuitive. There is even no name of the proposed framework.
2. Besides the a bit engineering method combinations, there seems to be no real algorithm novelty to the neural and behavior analysis community.
3. Some components of the proposed framework, like bi-directional RNNs, are actually a bit old-fashioned. The framework overall is actually looks like a theoretically unvalidated construction.

**Questions:**

I have no more questions, other concerns please see my weaknesses section.

---

> ### Author Response · Authors · 2025-11-22
>
> We thank the reviewer for their time and feedback, and for acknowledging that our paper is "concrete and easy-to-follow" and addresses "quite important" problems in neural data modeling. We address the reviewer's concerns regarding novelty, our methods, and theoretical validation in our point-by-point responses below, in light of which we hope that the reviewer would reconsider their score.
>
> ---
>
> > [W1] The overall framework in the submitted manuscripts is like a combination of the incremental model enhancement of two existing methods, which is pretty heuristic and intuitive. There is even no name of the proposed framework.
>
> We clarify that the framework we provide here is supported by extensive theoretical derivations (see extensive proofs in the manuscript and its appendices) and therefore is not heuristic. We view this work as the critical **non-causal completion** of the PSID framework, derived with full theoretical rigor. This is analogous to how **Kalman Smoothing** (e.g., RTS smoothing) is the rigorously derived non-causal extension of the **Kalman Filter**. Crucially, we not only provide comprehensive empirical validations in simulations that confirm ground truth performance (Figure 2) as well as in real data (Figure 3), but also provide rigorous proofs for our derivations (see also **new Appendices A.8.3, A.8.5, A.8.6**). While the resulting forward-backward architecture may seem conceptually intuitive in hindsight, once the derivation is complete, the method for learning the **optimal** filtering and smoothing **parameters** in the two-signal setting is not trivial and has not been shown before. Specifically, we prove that the two-signal setting makes key filtering parameters (like $K_f$) uniquely identifiable, which is not true in the standard one-signal setting.
>
> On the DPAD front, we show that the same proven architecture of forward-backward filtering, when applied to the DPAD framework, can naturally extend it to the smoothing application in the nonlinear setting. DPAD can implement PSID in the special linear case, and here we show that DPAD with smoothing is the natural nonlinear counterpart of PSID with smoothing. We also show that it outperforms the state-of-the-art (SOTA) performance in one dataset in the public NLB benchmark while also reaching near SOTA performance in other NLB datasets (Table 1).
>
> Taken together, the methods provided here are grounded in rigorous derivations that enable novel **optimal** filtering and smoothing in the two-signal setting – a capability that was not previously attainable. These methods are thus not combinations of incremental model enhancements. Indeed, other reviewers also characterize our theoretical derivations as “solid” (KZ2E) and “strong” (2Qcp) and our empirical validations as “comprehensive” (KZ2E) and “strong” (55zD).
>
> **Regarding naming:** We intentionally avoided coining a new name for our methods to emphasize that these are smoothing extensions of the existing PSID and DPAD frameworks, much like “Kalman Smoothing” (RTS smoothing) is to the "Kalman Filtering". By presenting our work as the filtering and smoothing extensions of PSID/DPAD, we aim to make the relationship between these methods clear and their applications easier to identify for the community.
>
> ---

---

> > ### Author Response · Authors · 2025-11-22
> >
> > > [W2] Besides the a bit engineering method combinations, there seems to be no real algorithm novelty to the neural and behavior analysis community.
> >
> > As we clarified above, there is significant theoretical and algorithmic novelty in our work and experimental results, which we believe directly benefit the neural and behavioral analysis community. These include several novel contributions, such as: (1) the formal observation that the two-signal setting makes key filtering parameters (like $K_f$) uniquely identifiable, which is not the case in the one-signal setting; (2) derivation of optimal filtering and smoothing for the two-signal problem and (3) the new algorithm to identify the optimal filtering gain $C_z K_f$ via RRR.
> >
> > We believe that the novel theoretical contributions of our work in extending PSID are of interest both in the general system identification and representation learning literature, as well as the neural and behavior analysis community. And the fact that our extension of nonlinear DPAD outperforms SOTA or achieves near-SOTA performance across the widely adopted public NLB benchmark is evidence that DPAD, too, would be of interest in that community.
> >
> > ---

---

> > > ### Author Response · Authors · 2025-11-22
> > >
> > > > [W3-a] Some components of the proposed framework, like bi-directional RNNs, are actually a bit old-fashioned.
> > >
> > > We would like to clarify that the bi-directional RNN is not introduced as an architectural novelty. Rather, it serves as a standard non-linear implementation of the forward-backward smoothing principle that we derive and demonstrate for linear PSID. We find it elegant that in the causal case, linearizing DPAD reduces to PSID; and now there is a similar connection in the non-causal smoothing versions of these algorithms as we demonstrate here.
> > >
> > > Finally, although model architectures like Transformers are newer, RNN-based models remain highly competitive on the Neural Latents Benchmark, where RNN-based models (e.g., AutoLFADS, Ctrl-TNDM, and the proposed DPAD with Smoothing) continue to be the SOTA/near-SOTA; and perform comparably to, or even outperform, transformer-based approaches (e.g., NDT and NDT3) (Table 1).
> > >
> > > ---
> > >
> > > > [W3-b] The framework overall is actually looks like a theoretically unvalidated construction.
> > >
> > > We again wish to emphasize that our framework is rigorously validated, both theoretically and empirically.
> > > * **Empirical Validation:** Figure 2 shows that PSID with Filtering and PSID with Smoothing match the ideal model performance. In addition, Figure 5 shows that our framework identifies ground truth filtering model parameters with asymptotically vanishing errors. In the manuscript, we will emphasize this in Sections 4.1 and 4.2 to explicitly highlight these empirical validations and make their connection to the theoretical derivations clearer.
> > > * **Theoretical Validation:** The optimality of both PSID with filtering and PSID with smoothing follows directly from the derivations presented in the main text. In response to Reviewer 2Qcp’s request for additional clarity, we now explicitly present the corresponding formal proofs and sketches of proofs in **new Appendices A.8.3, A.8.5, and A.8.6**, which provide a more complete theoretical account for our optimal results.
> > > ---

---

### Official Review · Reviewer_2Qcp · 2025-11-01

**Soundness:** 3
**Presentation:** 3
**Contribution:** 3
**Rating:** 6
**Confidence:** 4

**Summary:**

This paper develops methods for filtering and smoothing two-signal SSMs, i.e., where observable $y\_k$ and $z\_k$ depend on latent $x\_k$ and the goal is to predict or infer $z$ from $y$. The proposed methods extend linear (PSID) and nonlinear (DPAD) methods to smoothing using forward-backward approaches. For PSID, they also add a method for system identification for optimal filtering that learns only the part of the $x,y$ dynamics that is identifiable from $z$.

**Strengths:**

Strong theoretical grounding.
Good experimental results on synthetic data and on neural decoding tasks against established baselines.

**Weaknesses:**

The paper claims the extensions of PSID have certain optimality properties but this is not proved. It is claimed that solving eq 7 (setting aside sampling error from a finite training set) yields the optimal filter for predicting $z_k$ from $y\_{1:k}$, but this is not quite proved. A similar question applies to PSID with smoothing, but for that algorithm there is also the more primary question of whether it yields the optimal (i.e., Bayesian or minimum-variance unbiased) estimate of $z\_k$ when the system parameters are known. I expect this would be easy to prove and would be important to add to the paper.

**Questions:**

I work a lot with SSMs and Kalman-type methods but had not seen models where the system and observation noise are dependent. Can you add a brief explanation or motivating example for why we should expect such dependence?

I was unfamiliar with DPAD and wonder whether it could benefit from tracking uncertainty ($P$). If I understand correctly the only latent state in the RNN is $x_k$, which is a mean estimate, whereas optimal filtering also requires covariance. Can the model be extended in this way, or does it somehow track uncertainty as is?

eq 12a: $A$ should be $I$

Duplicate \labels for eqs 19 and 25?

---

> ### Author Response · Authors · 2025-11-22
>
> We thank the reviewer for their thoughtful and thorough review and suggestions, and for highlighting the paper’s strong theoretical grounding and the quality of the experimental results. We have addressed all comments and questions in the updated manuscript, and below.
>
> ---
>
> ### **[W1] More formal proof for PSID with filter (equation 7)**
>
> We had initially only alluded to the proof, but have now added an extensive **new Appendix A.8.3** with a formal proof for PSID with filtering, including why equation 7 asymptotically results in optimal (in the sense of linear minimum variance unbiased) filtered estimates. We also emphasize that our numerical results show the performance of PSID with filtering is similar to ideal Kalman filters with ground truth model parameters (Figure 2b).
>
> ---
>
> ### **[W2] More formal proof for PSID with smoothing**
>
> In the **new Appendix A.8.5**, we have added a sketch of the proof for why PSID with smoothing should asymptotically achieve ideal smoothing. We also emphasize that our numerical results show that the performance of PSID with smoothing is similar to ideal Kalman smoothers with ground truth model parameters (Figure 2c).
>
> ---
>
> ### **[W3] Optimality of PSID with smoothing given correct system parameters**
>
> This is a very important question. In **Appendix A.5**, we show how the optimal smoother (a.k.a. the Kalman smoother) can be formulated with a two-filter forward-backward formulation very similar to our method. So one could parameterize our forward and backward models as those in the Kalman to compute the same forward and backward states. We have now added **two new Appendix sections (A.5.1 and A.8.6)** to make this relationship more formal and make our minor deviations from it explicit. In these additions, we show that if the forward and backward PSID filters are set to the true system parameters, then the resulting forward-backward procedure achieves the optimal smoothed estimate, up to a single weighting simplification.

---

> > ### Author Response · Authors · 2025-11-22
> >
> > ### **[Q1] Allowing dependence of state-observation noises**
> >
> > This is a great question. First, we emphasize that we do not require dependence of the noises; rather, we simply support the more general case of correlated noises in our formulation, but of course, independent state-observation noises are also supported as a special case (i.e., the case of $S=0$).
> >
> > To see why allowing for this dependence is useful, we note that the general formulation is useful for models with latent states, with one key application being in subspace-based system identification methods like PSID. In fact, the more general formulation with nonzero state-observation noise correlation (i.e., defined as $S$ in equation 1) is the norm in the subspace identification domain (see for example, Van Overschee \& De Moor, 1996; Katayama, 2006; Anderson \& Moore, 2012). This is in contrast to expectation maximization (EM) methods, where $S$ is often assumed to be 0, and this assumption is enforced in the maximization step of each iteration by simply setting $S$ to 0. In contrast, subspace identification methods are not iterative and operate by finding the subspace of the latent states directly from observations (via a projection and SVD). As such, subspace methods simply find some equivalent formulation of the latent state space model that does not necessarily have $S=0$ (see also Faurre’s Theorem, discussed in Appendix A.3). If desired, one could attempt to find an equivalent model with $S=0$, which may exist, but finding it is often unnecessary given that the optimal Kalman filter for non-zero $S$ is known (Appendix A.2).
> >
> > ---
> >
> > ### **[Q2] How DPAD can account for uncertainty**
> >
> > This is a very interesting question that gets to a key feature of DPAD: using the predictor form representation of state space models. We have added a discussion on this in the **new Appendix A.9**. Briefly, DPAD does this by directly learning inference parameters that are dependent on the uncertainty in the training data, as we expand below.
> >
> > It is easiest to answer this question in the linear case, where we know that the optimal solution that accounts for uncertainty correctly is a Kalman filter. The DPAD paper (Sani et al., 2024) shows in the Extended Data Fig. 1 that for linear state-space models, i.e., when all parameters of DPAD are set to be linear matrix multiplications, DPAD indeed reaches the same performance as an ideal Kalman filter with true model parameters. So how does this happen? Note that the steady state Kalman filter recursions can be written as equation 3, which we repeat below for easier exposition:
> >
> > $\hat{x}_{k+1|k} = $
> >
> > $       \qquad\qquad     (A - K C_y ) \hat{x}_{k|k-1}+ K y_k $
> >
> >
> >
> > This simple equation considers uncertainty optimally because the fixed matrix **$K$**, which is the steady state Kalman gain, is a function of the noise/uncertainty covariances. This recursive equation is also the basis for the equivalent predictor form formulation for the state space model (equation 4), which is what DPAD model formulation (equation 10) reduces to if every parameter is set to be linear (i.e., matrix multiplication). Thus, DPAD can implement an optimal Kalman filter by learning the final steady state Kalman gain $K$ numerically via gradient descent. In other words, as the reviewer puts it, DPAD tracks uncertainty as is, by directly learning (via numerical optimization) the inference recursions that yield the best prediction (lower MSE) in the training data.
> >
> > ---
> >
> > ### **[Q3] The two latex errors**
> >
> > We are grateful that the reviewer caught these errors and have now fixed them.

---

> > > ### Comment · Reviewer_2Qcp · 2025-11-24
> > >
> > > Thanks for the clear explanations. Regarding the question of whether DPAD can track uncertainty, the mechanism you describe works only in the steady state, so it isn't so much tracking uncertainty as it is learning a fixed posterior covariance (by learning a fixed gain).

---

> > > > ### Author Response · Authors · 2025-11-25
> > > >
> > > > We thank the reviewer for this helpful clarification. The reviewer is correct that the mechanism we described in Appendix A.9 is only about implementing an optimal Kalman filter *at steady state*. We have added “at steady-state” to make this more explicit:
> > > >
> > > > >“DPAD can implement an optimal Kalman filter at **steady-state** by learning the final steady state Kalman gain $K$, along with the other parameters of the above equation, numerically via gradient descent.”
> > > >
> > > > Moreover, we have added a discussion and expanded Appendix A.9 to explain this limitation. We repeat this new Discussion here for ease of access:
> > > >
> > > > >“Even given enough training data and model capacity, DPAD can be optimal only if the optimal inference for the system that is being modeled can be written in the form of recursions given in equation 10. One example of this is the optimal Kalman filter for a linear system at steady-state. In the special case of linear modeling, DPAD’s model (equation 10) reduces to the predictor form representation of the state-space model (equation 4), which is based on a steady-state Kalman filter (equation 20). In this case, it is straightforward to see how DPAD can implement an optimal Kalman filter at steady-state, by directly learning all its steady-state parameters via gradient descent (Appendix A.9). Similarly, the bidirectional RNN in DPAD with smoothing can implement the operations of a Kalman smoother at steady-state (Appendix A.5.1). Finally, with additional latent states, the nonlinear RNNs in DPAD have the capacity to also learn time-varying model dynamics that can be formulated as equation 10, such as a state transition matrix that itself evolves over time with a random walk. DPAD could model such time-varying dynamics by dedicating some latent states to tracking the time-varying system parameters. However, deriving theoretical guarantees on optimality or bounds on errors in modeling such time-varying dynamics and for general nonlinear dynamics are beyond the scope of this work (Appendix A.9).”
> > > >
> > > >
> > > >
> > > > P.S. The reviewer did not ask for this, but we have also taken this opportunity to update our proof of PSID with filtering (Appendix A.8.3) to a more general version that removes the Gaussian assumption for $\epsilon_k$.

---

### Official Review · Reviewer_KZ2E · 2025-11-01

**Soundness:** 3
**Presentation:** 2
**Contribution:** 3
**Rating:** 6
**Confidence:** 2

**Summary:**

This paper extends PSID (linear) and DPAD (nonlinear) methods to support optimal filtering and smoothing of a secondary signal (e.g., behavior) from a primary neural signal. The authors derive theoretical updates, introduce a forward-backward smoothing framework, and validate the methods on both simulated and real neural datasets, showing improved decoding accuracy over existing baselines.

**Strengths:**

* Clear problem formulation for prediction, filtering, and smoothing in two-signal settings.
* Solid theoretical derivation for extending PSID to filtering via reduced-rank regression.
* Comprehensive evaluations on simulations, primate data, and NLB benchmarks.
* Unified framework covering both linear and nonlinear models for causal and non-causal inference.

**Weaknesses:**

* There seems to be no discussion on computational cost or real-time applicability.
* The current abstract is a bit long.

**Questions:**

* Whether the forward and backward RNNs can be replaced with GRU or LSTM, since they could provide more smoothness.
* What's the definition of $\\boldsymbol \\epsilon_k$? Does it have to be Gaussian?

---

> ### Author Response · Authors · 2025-11-22
>
> We thank the reviewer for their thoughtful feedback and suggestions. We are glad they found our problem formulation clear, the theoretical derivation solid, and the evaluations comprehensive. We have addressed their points in the revised manuscript.
>
> ---
>
> ### **[W1] Computational Cost and Real-time Applicability**
>
> This is a great suggestion. We have added the **new Appendix A.13** to discuss the computational cost for training and inference of each method in detail. Briefly, for both PSID and DPAD, the **inference cost** of the smoothing versions derived here is roughly twice that of their causal counterparts, because the exact inference operations happen twice, once in the forward direction and once in the backward direction. As a numerical example, on our desktop computer with an Intel Core i7-9700K @ $3.60\, \text{GHz}$ CPU, with a state dimension of $n_x=32$ on the NLB data, PSID takes $0.0061\ \text{ms}$ per time-step, PSID with Smoothing takes $0.0124\ \text{ms}$ per time-step, while causal DPAD takes $0.210\ \text{ms}$ per timestep, and DPAD with smoothing takes $0.254\ \text{ms}$ per time-step.
>
> In terms of the **computational cost of training**, for PSID, the cost is slightly more than doubled in the smoothing version, given the forward and backward application of PSID with smoothing, which includes the additional reduced-rank regression step. For DPAD, the computational cost of learning is not deterministic, given that the method uses a numerical optimization. But in our experiments, we found that under the exact same conditions (data size, state dimension, number of epochs, etc.), the average cost was 1.32 times higher for DPAD with smoothing compared to causal DPAD.
>
> Regarding **real-time applicability**, it is worth emphasizing that one of the key benefits of both PSID and DPAD is that they have computationally efficient inference and can easily run in real time due to the recurrent nature of their latent state estimation. This means that the smoothing versions derived here, which have roughly double the computational cost, are also computationally efficient. Nevertheless, these smoothing versions are inherently offline methods. This is not due to computational cost, but because by definition, smoothing requires access to future data. Nevertheless, in real-time applications where some delay (e.g., $d$ samples) is acceptable, one could still use these computationally efficient smoothing methods by processing the data in a sliding window and always returning the smoothed estimates from $d$ samples before the latest sample in the window – this is because even smoothing is highly computationally efficient, for example, $0.0124\ \text{ms}$ for PSID and $0.254\ \text{ms}$ per time-step for DPAD, which is much shorter than typical bins used to process neural data in brain-computer interfaces ($\sim 10-50\ \text{ms}$ bins).
>
> ---
>
> ### **[W2] Abstract Length**
>
> We have shortened the abstract in the new version of the manuscript, while still covering all key points.
>
> ---
>
> ### **[Q1] RNN vs. GRU/LSTM**
>
> This is an excellent question. As with DPAD, one of the design choices in our nonlinear models is the type of recurrent cell (symbolized by the $A’$ parameter), which could be set to an LSTM or GRU instead of a vanilla RNN (Section 3.5). In fact, in our NLB results, we were already using an LSTM (as noted in Appendix A.11). We have added a new **Figure 9** to the new **Appendix A.18**, which shows a direct comparison between vanilla RNN and LSTM cells on all 3 NLB datasets, confirming that LSTM cells achieve better results across most state dimensions.
>
> ---
>
> ### **[Q2] Definition of $\epsilon_k$**
>
> We have added a clarification for this in Section 3.1, where $\epsilon_k$ is defined. Briefly, we define $\epsilon_k$ as in PSID (Sani et al., 2021), as a general random process that represents the dynamics of the secondary signal ($z_k$) that are not encoded in the primary signal ($y_k$). As such, we do not need an assumption on its distribution; rather, we only assume that it is zero-mean and independent of $x_k^s$ and the other noises ($w_k$ and $v_k$).
>
> ---

---

### Author Response · Authors · 2025-12-03

We would like to thank all reviewers for their time and valuable feedback. As noted in our point-by-point responses, we have fully addressed all their comments and revised our manuscript according to their feedback. It is unfortunate that at this ICLR the discussion period closed early and before the reviewers had the chance to engage further, but we are hoping they are satisfied with our revised manuscript that is now stronger due to their feedback.

---

### Meta-Review · Area_Chair_2kcH · 2026-01-04

**Summary:**

This paper extends two existing prioritized two-signal dynamical modeling approaches, PSID for the linear case and DPAD for the nonlinear case, to support not only causal one-step-ahead prediction but also filtering and forward-backward smoothing of a secondary signal from a primary neural signal. The topic is relevant and the scope is broad, combining theory, simulations, and multiple real datasets including NLB. However, the decision is mainly driven by uneven reviewer reception and, more importantly, lingering doubts about how much of the work is genuinely new versus a fairly direct completion of existing frameworks, especially on the nonlinear side. While the authors emphasize optimality and identifiability results for the linear setting, one reviewer remained strongly unconvinced that the contributions rise above incremental extensions, and the paper’s presentation of the theoretical novelty was a consistent pain point. Given the split and the remaining uncertainty about novelty and positioning, I am leaning toward rejection.

**Reviewer Concerns:**

The rebuttal clearly helped on the “easy-to-fix” issues. The authors added substantial material clarifying computational cost and practical implications, expanded formal proofs for the PSID filtering and smoothing claims, and improved structure by adding more theorem-like statements and appendices. For the positive reviewers, these responses likely address their main requests.

That said, the core concern from the negative review was not really about missing details. It was about whether the paper’s contributions are fundamentally new or largely a combination of straightforward extensions to known methods. The authors argue that the two-signal setting yields additional identifiability and that learning the filtering gain through reduced-rank regression is nontrivial, and they also position the bidirectional DPAD as the nonlinear counterpart of the forward-backward construction. Even if those points are correct, the paper still reads, to at least one reviewer, as a “completion” of PSID and a fairly standard bidirectional-RNN smoothing wrapper for DPAD, without a strong enough conceptual or algorithmic leap to justify acceptance at ICLR. In other words, the rebuttal strengthens the rigor, but it may not change minds on the central novelty question.

**Reviewer Scores:**

Since reviewers weren’t allowed to update scores this year, I can only infer how things would have moved after the rebuttal. The three reviewers who were already positive (all at 6) would probably stay in the same place, just with a bit more confidence now that the authors added the missing proofs, computational cost discussion, and other clarifications. The real question is whether the reviewer who gave a 2 would budge. My guess is the rebuttal might pull that score up somewhat if they go back and read the theoretical sections more carefully, especially with the new formal material. But I’m not convinced it would be enough to flip them into an accept, because their main objection wasn’t “the math is unclear”, it was “this feels incremental and not truly novel.” Overall, I’d expect the gap to narrow a little, but the reviews would still look mixed, and not strong enough for me to recommend acceptance.

---

### Decision · Program_Chairs · 2026-01-26

Reject